# Stream nitrate enrichment and increased light yet no algal response following forest harvest and experimental manipulation of headwater riparian zones

Sherri L. Johnson[1]*, Alba Argerich[2], Linda R. Ashkenas[3], Rebecca J. Bixby[4], David C. Plaehn[5]

1 Pacific Northwest Research Station, USDA Forest Service Corvallis, Corvallis, Oregon, United States of America, 2 School of Natural Resources, University of Missouri, Columbia, Missouri, United States of America, 3 Department of Fisheries and Wildlife, Oregon State University, Corvallis, Oregon, United States of America, 4 Department of Biology, University of New Mexico, Albuquerque, New Mexico, United States of America, 5 Pilot Records, Corvallis, Oregon, United States of America

* sherri.johnson2@usda.gov

**Data Availability Statement:** These data are archived and publicly available through the US Forest Service Research Data Archive. Data and

## Abstract

Disturbances to forested watersheds often result in increases of nutrients and light to nearby streams. Such changes are generally expected to produce a shift to a more autotrophic aquatic ecosystem, with measurable increases in algae, and associated implications for food webs and fisheries. Although this paradigm is widely established, results from our 10-year study (2007–2016) in 12 headwater streams and four sites downstream in the Trask River Watershed (Oregon, USA), did not concur. In 2012, one watershed was thinned, three were clearcut harvested with variable buffers and three with uniform riparian buffers. After harvest, light to the stream surface significantly increased at the three watersheds with variable buffers while dissolved inorganic nitrogen (DIN) significantly increased in all of the clearcut harvested streams. Despite the increase in DIN and light, algal standing stocks and chlorophyll *a* concentrations did not significantly increase. The common assumption of increased autotrophic responses in stream food webs following increases of nitrogen and light was not supported here. We postulate the co-limitation of nutrients, driven by low phosphorus concentrations, which unlike DIN did not increase post-harvest, and the characteristics of the algal community, which were dominated by low light adapted diatoms rather than green algae, contributed to our findings of no responses for standing stocks of epilithic algae or concentrations of chlorophyll *a*. The inclusion of multiple statistical analyses provided more certainty around our findings. This study documents responses to current forest practices and provides cautionary information for management and restoration activities aiming to increase fish abundance and standing stocks by opening riparian canopies and adding nutrients.

supporting documents are online and cited in the bibliography of the manuscript. https://doi.org/10.2737/RDS-2022-0001 https://doi.org/10.2737/RDS-2022-0002 https://doi.org/10.2737/RDS-2022-0003.

**Funding:** Support for the Trask River Watershed Study was primarily provided through the Oregon State University Watershed Research Cooperative, with long-term base funding from the Oregon Department of Forestry and Weyerhaeuser Corporation. Additional funding was provided by US Bureau of Land Management, US Forest Service Pacific Northwest Research Station, US Geological Survey Forest and Rangeland Ecosystem Science Center, Oregon Department of Environmental Quality, Oregon Watershed Enhancement Board. The funders provided support in the form of salaries for authors [SLJ, AA, LRA, RJB], but did not have any additional roles in the decisions about data collection, data analysis and interpretation, preparation of the manuscript and decision to publish.

**Competing interests:** The authors have declared that no competing interests exist.

# Introduction

## Terrestrial disturbances in headwaters and the role of riparian forests in maintaining stream integrity

Researchers have long worked to understand instream responses associated with differing types of watershed disturbances, including forest harvest [1, 2], fire [3, 4], hurricanes [5], agriculture [6], and urbanization [7]. Forests and woodlands cover large portions of landscapes; across the Unites States, they comprise one-third of the land base [8]. Forested watersheds managed for timber production (including harvest) some 67% of the forested lands in the USA [8]. Evaluation of the effects of forest harvest on streams ecosystems is therefore an important and necessary focus of research and monitoring [9]. Streams flowing through forests provide habitat for a diverse array of aquatic organisms [10, 11] as well as serve as sources of drinking water for downstream communities [12]. Because headwater streams represent 80–90% of miles of any given river network [13], reducing impacts from forest harvest to headwater streams can have broad implications locally as well as downstream [11, 14].

Forested riparian zones provide a variety of functions [15, 16] including buffering aquatic ecosystems from disturbances within their watersheds [9, 17]. Intact riparian vegetation can stabilize stream banks [18], provide shade to maintain cool water and air temperatures [19, 20], minimize direct solar radiation that can potentially stimulate algal blooms [2, 21], intercept nutrients and sediment transported from hillslopes [22, 23], and deliver allochthonous detrital material, including wood, leaves, and needles [24, 25]. Changes to riparian vegetation and/or the widths of riparian buffers during natural or anthropogenic disturbances can result in shifts in some or all of these functions, potentially leading to impacts on multiple trophic levels of instream biotic communities [26–29] and water quality for downstream water users [9, 30].

## Management of riparian forests to reduce instream responses to harvest

The assumptions and observations of instream responses to riparian management are numerous and sometimes conflicting, which can make it challenging to identify best management practices and regulatory guidelines during forest harvest. Identifying the optimal widths of riparian buffers necessary for protection of instream functions is complicated by multiple site-specific factors capable of influencing riparian-stream interactions [23, 31] as well as which stream response parameter or process is of most interest [17, 32, 33]. For example, headwater forested streams frequently have low background levels of stream nutrients [34, 35] and following harvest in forested watersheds, increases in dissolved nitrogen, phosphorus and carbon have been observed [36–40]. Increases in stream chemistry concentrations have been associated with increased instream sediment [29, 39, 40], especially when riparian buffers were not retained during harvest. However, not all studies have observed increased stream nutrients or stream sediment following forest harvest [35, 41, 42]. Retention of vegetated riparian buffers during harvest has been suggested as a way to mitigate these increases [16, 43, 44].

The removal of riparian vegetation during harvest has also been frequently associated with increases in stream algae [45, 46]. Riparian harvest, and the accompanying changes in instream light and possibly nutrients [45, 47, 48] can result in higher biomass of attached algae and increased concentrations of chlorophyll *a* [49, 50]. In studies of forest streams, an increase in algae has been linked to higher densities of macroinvertebrates and fish [47, 49, 51–53]. Depending on site-specific interactions, increases in nutrients and potential co-limitation between nitrogen and phosphorus can influence algal biomass [54, 55]. The taxonomic composition of algal assemblages can shift in response to changes in light and nutrients. Several

studies have documented a shift from diatom communities to filamentous green algae follow-ing clearcutting [45, 46, 56]. Although the relationships among reduced riparian cover, increased light and increased algal standing stocks are frequently assumed to be ubiquitous [21, 57], a subset of research papers have noted that algal responses to increased light and nutrients can be negligible [46, 58, 59]. It has been suggested that null findings that do not sup-port this common paradigm are less frequently published [58]. Publications of null findings are valuable because they influence the background information available for new studies as well as document exceptions to assumptions attributed to instream responses to forest management.

Compounding the complexity of instream responses to forest harvest is the inflexibility of commonly used statistical tests to capture biological variability and statistical significance in dynamic systems such as streams. Traditional statistical analyses have generally been devel-oped for controlled studies with high replication and constant variance. The statistical assump-tions for parametric analyses of normally distributed, homoscedastic data, randomly assigned treatments and independent measurements may not be appropriate for field studies of land-scape-scale experiments with low sample sizes. Yet the statistical significance associated with natural resource questions is highly relevant to forest planning and policy, which has led to multiple instances of controversies about the best statistics [60–63]. Given the potential com-plexity of types of statistical assessments, research that informs best management practices, including management of riparian vegetation during harvest, would ideally address these mul-tifaceted dynamics with multiple, rigorous evaluations of responses to current forest harvest practices.

In this landscape-scale experimental study, we use multiple statistical tests to examine whether differences in the width of riparian buffers retained during clearcut harvest preserve water quality and whether stream ecosystems shift to include more autochthonous resources. Our hypotheses were that removal of some or all riparian vegetation would lead to a) higher concentrations of instream nutrients and dissolved organic carbon, b) higher chlorophyll *a* concentration and algal biomass in response to more available light and nutrients, and c) increased instream sediment as a result of disturbances to hillslope soils from harvest activities. We also hypothesized that responses within a harvested area might propagate to downstream sites.

## Methods

### Site description and experimental design

Study sites were located in the Trask River Watershed Study area, within the upper 25 km$^2$ of the South Fork of the Trask River in Oregon, USA (Fig 1). The Trask River originates in the northern Oregon Coast Range and flows westward into the Pacific Ocean. The underlying geology is a mixture of sedimentary deposits and volcanic basalt intrusions [64]. Annual rain-fall in the upper Trask study area averages 200 cm per year, primarily between October and May, with occasional snow at higher elevation sites. At the beginning of this study, the hillslope and riparian forests in the Trask River Watershed Study area were 45–60 year old Douglas-fir (*Pseudotsuga menziesii*) and red alder (*Alnus rubra*). Big leaf maple (*Acer macrophyllum*) occurred only in riparian areas. This landscape had been heavily impacted by prior distur-bances; a series of wildfires occurred between 1933 and 1951 and following the fires, all burned and unburned trees were harvested and the area replanted. In addition to impacts from fires and harvest, land management legacies included a high density of logging roads, minimal downed wood in streams and riparian areas, and a simplified understory plant community.

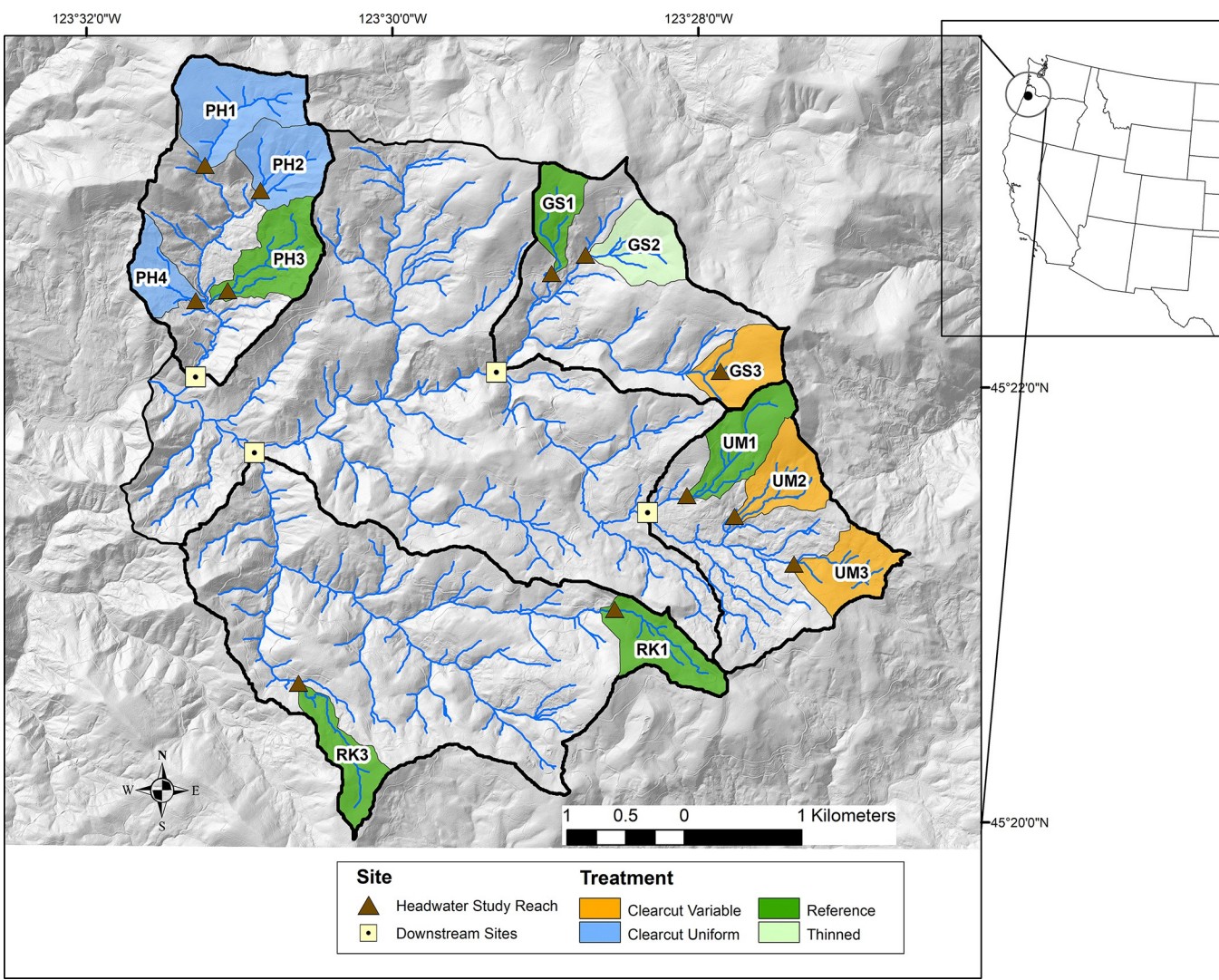

**Fig 1. Map of study watersheds in Trask River Watershed Study, western Oregon.** Locations of headwater sampling reaches are shown as triangles and downstream sites are noted as squares. Map was created using ArcGIS Pro with LiDAR terrain (https://www.oregongeology.org/lidar/) and derived hydrography. Reference watersheds are shown in dark green, clearcut with variable buffers are shown as orange and clearcut with uniform buffers are show as blue. Light green watershed was thinned with wide unharvested buffers. Created using ArcGIS Pro with LiDAR terrain (https://www.oregongeology.org/lidar/) and LiDAR-derived hydrography.

The Trask River Watershed Study was conducted by researchers and natural resource managers as a multi-disciplinary, 10+ yr. collaborative study to understand and quantify the onsite effects of current timber harvest practices on physical, chemical, and biological communities in first-order headwater streams. We also evaluated whether responses to headwater harvest propagated downstream to third-order fish-bearing streams. This study was conducted using a paired study design of 12 small watersheds nested within four large catchments (Fig 1). Within each large catchment, two to three small headwater watersheds (17–67 ha) were clearcut harvested over as much of the watershed area as possible, while one headwater watershed remained as an unharvested reference site (Table 1). In addition, no harvests occurred in one of the larger catchments (Rock Creek) so that it could serve as a reference site for the downstream comparisons. The pre-harvest sampling occurred 2007–2011. In 2012, harvest took

**Table 1. Study site characteristics.**

| Catchment | Watershed | Harvest treatment [1] | Ownership | Elevation Study Site (m) | WS Area (ha) | Percent WS harvested in 2012 | Distance to downstream site (m) | Ave channel slope (%) | Ave wetted channel width (m) | Ave water depth (cm) | Ave discharge (L/s)[2] |
|---|---|---|---|---|---|---|---|---|---|---|---|
| **Upper Main** | UM 1 | Reference | Private | 658 | 44.2 | 0% | 406 | 18.7 | 1.47 | 6.8 | 6.6 |
| | UM 2 | Clearcut variable buffer | Private | 685 | 17.1 | 91% | 1140 | 18.4 | 1.16 | 5.8 | 5.9 |
| | UM 3 | Clearcut variable buffer | Private | 728 | 39.2 | 56%* | 1581 | 11.8 | 1.34 | 6.9 | 5.2 |
| | UM Downstream | Downstream of harvested | Private | 608 | 278.8 | 25% | – | 3.8 | 3.63 | 14.1 | 15.1 |
| **Gus** | GS 1 | Reference | Private | 611 | 26.5 | 0% | 1059 | 12.1 | 1.32 | 6.1 | 3.7 |
| | GS 2 | Thinned | Federal | 638 | 39.0 | 91% | 1355 | 14.5 | 1.82 | 7.0 | 5.6 |
| | GS 3 | Clearcut variable buffer | Private | 789 | 20.4 | 92% | 2297 | 22.4 | 1.20 | 4.2 | 1.2 |
| | GS Downstream | Downstream of harvested | State | 470 | 301.4 | 30% | – | 10.2 | 4.16 | 15.4 | 27.6 |
| **Pothole** | PH 1 | Clearcut uniform buffer | State | 531 | 67.1 | 87% | 2352 | 13.6 | 2.00 | 10.1 | - |
| | PH 2 | Clearcut uniform buffer | State | 493 | 20.4 | 82% | 2159 | 13.7 | 1.00 | 6.2 | 4.9 |
| | PH 3 | Reference | State | 390 | 45.4 | 0% | 1044 | 12.8 | 0.93 | 5.5 | 4.3 |
| | PH 4 | Clearcut uniform buffer | State | 370 | 23.6 | 92% | 947 | 13.8 | 0.80 | 5.2 | 1.5 |
| | PH Downstream | Downstream of harvested | State | 324 | 324.6 | 44% | – | 2.8 | 3.30 | 13.0 | 19.4 |
| **Rock** | RK 1 | Reference | Private | 680 | 42.2 | 0% | 4385 | 7.6 | 1.36 | 9.5 | 11.7 |
| | RK 3 | Reference | State | 548 | 35.3 | 0% | 2388 | 16.2 | 1.66 | 9.9 | 12.3 |
| | RK Downstream | Downstream reference | State | 334 | 669.9 | 0% | – | 3.5 | 4.58 | 16.3 | 43.1 |

**[1] Harvest details:**

Clearcut variable buffer: A riparian buffer was not required on these sites. During harvest, leave trees remained in the downstream riparian areas of UM2 and UM3

Clearcut uniform buffer: Clearcut or modified clearcut with a uniform 12 m riparian buffers on each side of the stream

Thinned: Thinned watershed with a 15 m no-harvest riparian buffer on each side of the stream

Reference: Not harvested during this study period. Vegetation age was 45–60 yrs.

[2] Discharge was measured with conservative salt tracers at the same time as the water chemistry and epilithon sampling.

* A portion of this WS had been harvested 6 years previously

place in seven headwater watersheds and no sampling occurred. Post-harvest, sampling restarted in 2013 and continued through 2016.

Riparian treatments during forest harvest differed by land ownership (Fig 1; Table 1). On private forest lands, riparian buffers were not required at the time of this study [65, 66] and the watersheds owned by forest industry (UM3, UM2, and GS3), were clearcut with variable amounts of riparian vegetation retained. Forest harvest guidelines for protection of terrestrial wildlife require the equivalent of two live trees per acre not be harvested [65]. In two watersheds (UM2 and UM3), the trees designated for wildlife were grouped in the riparian areas for operational reasons including to avoid steep banks or wet soils. We termed this treatment group as "Clearcut variable".

During harvest on State Forest lands (PH1, PH2, PH4), the ODF Forest Management Plan required minimum 7.6 m buffers along each side of the perennial section of the stream channel

[67]. In addition, some of the required trees designated for wildlife were grouped adjacent to the buffers which led to final buffers of approximately 12 m [19, 68]. We characterized these sites as clearcut with uniform width buffers, and termed this treatment group as "Clearcut uniform".

The thinned watershed (GS2) was on federal land, managed by the Bureau of Land Management. Harvest practices involved thinning throughout the watershed using helicopter logging and retaining a 15 m, no-entry buffer on each side of the stream. This site is termed thinned with uniform buffer and abbreviated as "Thinned". No additional forest treatments beyond these described occurred during the study period. More information is available as supplementary material with the archived data.

The locations of the study reaches were designed to spatially coincide with collaborative studies, which included stream temperature [19, 69], suspended sediment [64, 70], stream chemistry during storms [71], stream discharge, macroinvertebrates [68], amphibians [72] and fish [73]. In the harvested watersheds, the study reaches were located within the downstream most portion of the treated area. In addition, to assess potential cumulative impacts of harvest in upstream headwaters, four downstream study reaches were located in larger third-order fish-bearing streams, one in each of the catchments (Fig 1). Three study reaches were downstream of harvested watersheds, at distances of 947–2297 m (Table 1): UMDS was downstream of sites clearcut with variable buffers, PHDS was downstream of sites clearcut with uniform buffers and GSDS had mixture of harvest groups upstream. Rock Creek, RKDS, was the reference for the downstream sites.

Each year, the sampling for the research occurred in early summer to provide as coincident measurements of concentration, biomass, or abundance as possible for all of the study groups. In early summer, streamflow was sufficient at all headwater sites for consistent biotic sampling to occur. Sampling schedules were established to prevent unintended impacts of each type of sampling on subsequent projects. The Trask headwater sites did not contain fish, but all were inhabited by an amphibian predator, Coastal Giant salamander (*Dicamptodon tenebrosus*), and a subset of the sites contained an amphibian grazer, Coastal Tailed Frog tadpoles (*Ascaphus truei)*. At downstream sites, resident fish species included Cutthroat Trout (*Onchorhynchus clarkia clarkii*) and Sculpin (*Cottus* spp.). Two of the downstream sites also contained anadromous Coho Salmon (*Oncorhynchus kisutch*) and Steelhead (*Onchorhynchus mykiss*) on a seasonal basis.

## Data collection

**Stream characterization.** Sampling occurred annually during the last two weeks of June between 2007–2011 (pre-harvest period) and 2013–2016 (post-harvest period). On each sampling date, at each of the 12 headwaters and 4 downstream sampling sites, we collected basic measurements of stream physical characteristics. Channel dimensions were measured at a series of 4-transects over a 30 m reach in headwaters and at 10 transects over 100 m reach at downstream sites. Discharge at the time of sampling was calculated using a constant addition of a conservative salt tracer [74]. We avoided sampling during rain events. Water temperatures were measured continuously from June-September using recording thermistors (Onset Computer Corporation, model U22-001) protected from direct radiation [19]. Reach slope was measured in the field using a stadia rod and clinometer (PM-5, Suunto) in 2008 and 2013. At each sampling time, the average percentage of canopy closure was measured at each transect from the center of the stream using a handheld concave densiometer [75]. In addition, in 2008 (pre-harvest) and 2013 (immediately after harvest), we quantified the amount of overstory light reaching the stream using hemispherical photos. The hemispherical photos were

recorded using a self-leveling camera with hemispherical lens that was positioned mid-channel and 1m above the water surface. A total of five photos were taken per reach with photos spaced 25 m apart in headwater channels and 50 m apart at downstream sites. Photos were taken at dawn or dusk and when there were no clouds to minimize reflection or sunspots that can bias calculations of incoming light and canopy cover. In 2008, the position of camera point was carefully documented so that post-harvest images could be collected from the same position and height. The percent of overstory light reaching the stream was calculated for each image using a digital image analyzer (Metric used: percent total site factor; WinSCANOPY, Regent Instruments, Inc.) and the site average calculated.

**Water chemistry.** At each headwater and downstream site during the annual June sampling, we collected a grab sample of stream water for analyses of water chemistry. The samples were stored on ice in the field, and then filtered through pre-combusted Whatman GF/F filters (pore size = 0.7μm) in the laboratory within 3 days of collection. After filtering, the samples were frozen for later analysis at the Oregon State University Cooperative Chemical Analytical Laboratory (see http://ccal.oregonstate.edu/methodology for description of analytical methods and detection limits). Analytes reported here include dissolved organic carbon (DOC), soluble phosphorus (SRP), dissolved inorganic nitrogen (DIN; sum of nitrate, nitrite, and ammonium), dissolved organic nitrogen (DON; calculated as the difference between total dissolved nitrogen and DIN), total dissolved nitrogen (TDN) and total dissolved phosphorus (TDP).

**Algae.** Each year in late June, four composite samples of epilithic algae, hereafter termed epilithon, were collected at each of the 12 headwater sites and six composite samples were collected at each of the four downstream sites. Epilithon is a complex matrix of algal cells on stream substrates; it potentially includes bacteria, fungi, and fine detrital matter. The epilithon samples were analyzed for photosynthetic pigments (chlorophyll *a* concentration) and standing stock (biomass). For each composite sample, we used a small wire brush to scrub a known area of randomly selected rocks from riffles, glides, and cascades, but not pools. In these headwater streams, pools are not consistent or common, and they are depositional areas where fine sediment accumulations might affect epilithon standing stock. Each composite contained material scrubbed from four to twelve rocks, with a total sample area of at least 50 cm$^2$. Each rock was gently lifted out of the water and a flexible plastic template with a known area was held onto the rock, exposing only the area to be scrubbed. After scrubbing, the loosened epilithic material within the template area was gently rinsed into the sample container. The total area scrubbed from multiple rocks was recorded for each composite sample. Sample containers were placed on dry ice in the field, frozen, and analyzed within one month of collection. In 2016, additional sampling for epilithon and chlorophyll *a* was conducted in late July at sites with variable buffer, uniform buffer, and a subset of reference sites.

In the lab, each sample was split into two aliquots after thawing; the aliquot for epilithon biomass, as ash free dry mass (AFDM), was sieved, using 2 mm mesh, to remove large, non-algal particles, and filtered onto pre-combusted and pre-weighed GF/F filters (Whatman glass microfiber filters). These filters were then dried at 60˚C, weighed, combusted at 500˚C, and reweighed. The second aliquot was analyzed for chlorophyll *a* and phaeophytin concentration by filtering a known volume onto pre-combusted GF/F filters, and using the heated ethanol methodology for pigment extraction [76]. All measurement of concentrations of pigments occurred using a Shimadzu UV-1700 UV-VIS spectrophotometer. Chlorophyll *a* pigment concentrations were corrected for acid strength and post-acidification time [77].

We also collected algal samples for analysis of algal taxonomic structure from sites. At each sampling time, a fifth composite sample was collected as described above and preserved in 0.5% formalin. Samples from 2011 (pre-harvest) and 2013 (post-harvest) at a subset of sites were selected for analysis. The dominant algal genera were qualitatively noted in wet mounts

in a Palmer Maloney chamber [78]. To determine diatom densities, 4-ml aliquots were heated with 25 ml of 30% hydrogen peroxide for one hour and rinsed six times with distilled water to remove oxidation by-products. Processed subsamples were evaporated onto coverslips and mounted to microscope slides with Z-Rax mounting medium, making permanent slides. Specimens along transects were examined under oil immersion at 1000× magnification using a Leica DM5000 compound microscope with differential inference contrast microscopy. Five hundred valves were identified to species and enumerated from each sample. Diatoms were identified to the lowest taxonomic level (usually species), using standard taxonomic literature (https://diatoms.org/) and regional literature [79, 80]. Identifiable fragments were mathematically reconstituted to whole valve units when possible and included in total counts. The raw data were converted to densities and classified by growth forms, based on Rimet and Bouchez [81].

**Benthic sediment.** Each sampling period, three randomly located samples of fine benthic sediment were collected within the 30 m headwater sites following LINX II protocols [82]. Five samples were collected from the downstream sites. This benthic sediment sampling was sequenced before the algal sampling to avoid substrate disturbance and resuspension of sediment during other instream sampling. After collection, samples were refrigerated and returned to the laboratory. Sediment samples were dried and weighed for calculations of mass of fine surface sediment per square meter.

## Data analyses

We compared multiple statistical approaches to examine whether forest harvest treatments resulted in differences in responses between the pre- and post-harvest periods. We grouped our headwater study sites by four treatments–clearcut with variable buffer width (variable buffer; 0–7.5 m), clearcut with uniform buffer width (uniform buffer; 12 m), thinned with uniform buffer width (thinned; 15 m), and reference (no harvest during this study). Downstream sites were characterized into two groups based on whether there was harvest in the headwaters or no harvest. As well as examining treatment groups, we also modelled differences in responses between pre- and post-harvest periods at individual headwater sites. Our analyses and models examined responses of chlorophyll *a* concentration, epilithon biomass, surface sediment, concentrations of DOC, DON, DIN, and SRP, DIN:SRP molar ratio, and canopy cover following treatments relative to pre-harvest. We also calculated Pearson correlations between epilithon, chlorophyll *a*, canopy closure, DIN and molar ratio of DIN:SRP. A significance level of 0.05 was used for all tests.

**Kolmogorov-Smirnov test and bootstrapped means.** An initial step was to compare the pre- and post-harvest cumulative distribution functions for each treatment group (clearcut variable, clearcut uniform, thinned, and reference) using the non-parametric Kolmogorov-Smirnov 2-sample test (KS2) [83]. This test identifies whether the distributions are different, but does not provide information about direction of difference. We next evaluated whether the 95% confidence intervals (CI) for the means for pre- and post-harvest responses within a treatment overlapped using a bias-corrected, accelerated (BCa) bootstrap [84]. We considered pre- and post-harvest means significantly different if their bootstrapped confidence intervals did not overlap and noted the direction of the post-harvest change. These analyses were only conducted for treatment groups, because sample sizes for individual watershed sites pre- and post-harvest were small.

**Partial Least Squares and significance testing.** Our third approach involved modeling the differences in responses between pre- and post-harvest periods using Partial Least Squares regression (PLS) [85] to further evaluate differences in instream responses. Partial Least Squares is a latent variable approach that maximizes regressor and regressand covariance and

avoids problems with multi-collinearity. It creates multiple models for a given data set using latent variables, which can be thought of as similar to principal components. The criterion for selecting the "optimal" PLS model, essentially a notable change in slope of the root-mean-square-error of cross-validation (RMSECV) graph, which is usually at or near the minimum RMSECV (maximum $R^2CV$). If the $R^2CV$ of a given model was negative or "small," it was not used regardless of the $R^2$. Full (leave-one-out) cross-validation was used in the PLS modeling. Cross-validation allows for calculation of the $R^2$ of cross-validation ($R^2CV$), which we interpret as a more robust measure of model quality [86, 87].

Categorical factors of treatment group and harvest period, or watershed and harvest period were converted to dummy variables and the last level then deleted to avoid linearly dependent regressors. The interactions were then created from these dummy variables. For the model examining effects of forest harvest treatments, Response = Treatment + Period + Treatment*Period, where "Period" refers to harvest pre-post and "*" indicates the interaction between factors. This resulted in a regressor array with three Treatments, one Period and three interactions and allowed seven PLS latent variables, and therefore, seven regression models to be created. The downstream model was the same symbolically as the headwater treatment model except that there were only two categories in the treatment factor. For the headwater watershed model (Response = Watershed + Period + Watershed*Period), the regressor array had eleven Watersheds, one Period and eleven Watershed-Period interactions.

We used Box-Cox transformations [88] to try to improve model quality as defined by $R^2CV$ (S1 File). The $R^2CV$ of the resulting models was typically better than those of models that did not use transformed responses or more standard transformations, such as the log transformation.

In a standard regression model, one would generally assume the sites were independent of one another. In this study, this assumption was likely violated due to annual sampling at each site as well as the spatial distribution of treatments across watersheds which was a function of forest ownership. To address this concern, we employed Wu's jackknife [89] and the wild bootstrap [90, 91] for significance testing. For Wu's jackknife, we used one of Wu's best performing methods ("full cross validation"). This provided a coefficient variance estimate, which was then used in a 2-sided t-test for each coefficient for each model. We also used the wild bootstrap method $HC_3$ to construct 95% confidence intervals from 2000 re-samplings. The wild bootstrap confidence intervals were used to compare model pre- and post- predictions, where two predictions were considered significantly different if their 95% confidence intervals did not intersect. For regression coefficient testing, if a given confidence interval did not include zero, then the null hypothesis (that the coefficient was zero) was rejected. Note that neither the semi-parametric Wu jackknife nor the non-parametric wild bootstrap require normally distributed model residuals.

For calculations involving degrees of freedom for PLS models, the method described in Krämer and Sugiyama [92] was employed. Both the regressor and response data were centered and scaled. All analyses were conducted in Matlab™ 2020b (Mathworks, 2020) using Matlab programs, except Box-Cox, genetic algorithm, wild bootstrap, and Wu jackknife analyses, which were written by D. Plaehn.

## Results

### Stream characterization

Headwater streams were steep (mean slope 14.6%) and shallow (mean depth 6.9 cm), with wetted widths ranging from 0.8 m to 2.0 m (Table 1). Downstream sites had lower gradients (mean slope 5.1%), and were wider (mean width 3.9 m) and deeper (mean depth 15.0 cm). At

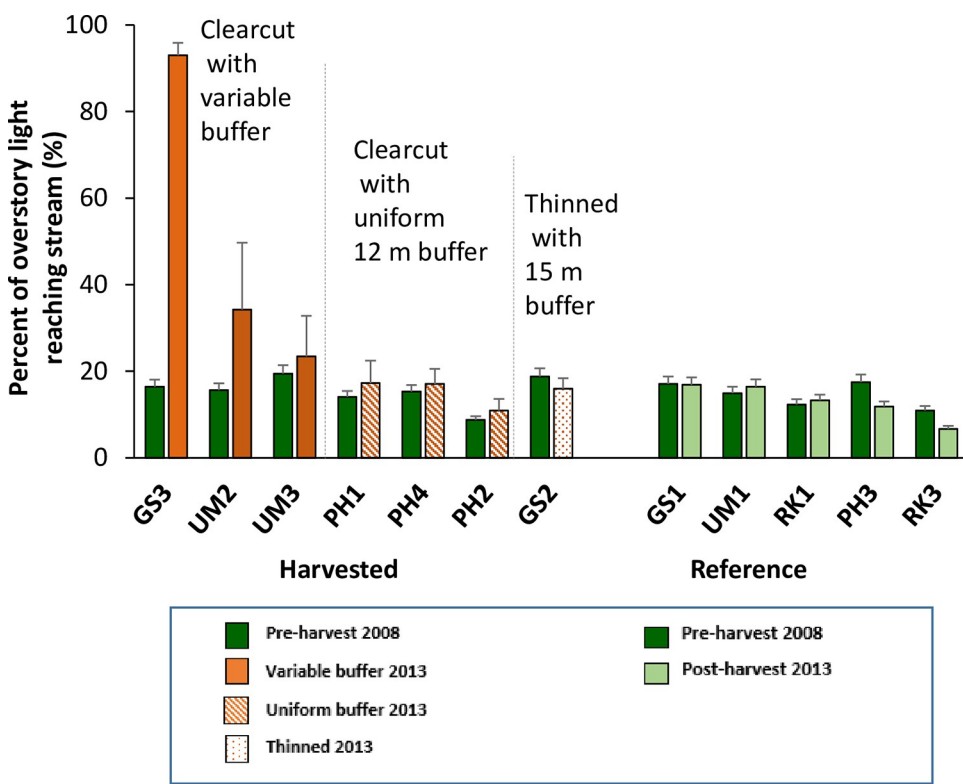

**Fig 2. Percent of light reaching streams pre- and post-harvest.** The percent of overstory light reaching the streams, measured using hemispherical photos in 2008, pre-harvest, and in 2013, post-harvest. Sites are organized from left to right by percentage of overstory light reaching the stream post-harvest. Bars show site averages and standard deviation.

the time of sampling, streamflow ranged from 1.2 L/s to 6.6 L/s, in headwater streams, with slightly higher discharge in the two reference watersheds in Rock Creek, whereas downstream sites showed larger discharge (ranging from 15–43 L/s). Streamflow across years was fairly constant during the three weeks preceding sampling with an average coefficient of variation of 22% among watersheds and years.

During the pre-harvest period, the percent of overstory light reaching the streams was low (average 15.5%) and comparatively uniform across sites (range 9–19%, Fig 2). After harvest in 2013, sites clearcut with variable buffers showed increased light reaching the stream. The annual measurements of percent canopy closure collected with a densiometer showed that canopy closure at clearcut sites with variable buffers was significantly different pre- and post-harvest in all three statistical approaches (Table 2, S1 Table). The three individual watersheds in the variable buffer group each showed significant changes in canopy closure pre- to post-harvest (Table 3). We did not observe significant changes in canopy closure for other treatments or for other individual watersheds.

Pre-harvest, maximum June water temperatures ranged between 7.3˚C and 15.3˚C in headwater streams. In the post-harvest period, maximum June temperatures were warmer than pre-harvest across all sites. Streams clearcut with variable buffers showed greater increases (3.9˚C, 2.5˚C, and 3.1˚C) than streams with uniform buffers (1.1˚C, 0.7˚C, 1.0˚C), thinned (1.2˚C) or reference streams (0.4˚C, 1.1˚C, 0.9˚C, 0.7˚C, 1.4˚C) (Fig 3A). In mid-summer, post-harvest temperatures at streams that were clearcut with variable buffers were significantly warmer than pre-harvest [19].

**Table 2. Statistical differences between pre- and post-harvest periods for headwater treatment groups.**

| Treatment group | Canopy closure | DOC | SRP | DON | DIN | DIN:SRP Molar Ratio | Epilithon biomass | Chl a | Surface sediment |
|---|---|---|---|---|---|---|---|---|---|
| **A. Kolmogorov-Smirnov two sample test p-values** | | | | | | | | | |
| Clearcut variable | **0.00*** | 0.43 | 0.43 | 0.99 | **0.00*** | **0.02*** | 0.98 | 0.92 | 1.00 |
| Clearcut uniform | 0.57 | 0.86 | 0.96 | 0.24 | **0.00*** | **0.00*** | 0.17 | **0.05*** | 0.54 |
| Thinned | 0.48 | 1.00 | 0.11 | 0.53 | 0.11 | 0.11 | 0.26 | 0.88 | 0.26 |
| Reference | 0.77 | 0.50 | 0.77 | 0.28 | 0.50 | 0.77 | **0.04*** | 0.06 | 0.38 |
| **B. Comparison of 95% BCa Bootstrap confidence intervals** | | | | | | | | | |
| Clearcut variable | **decrease** | 0 | 0 | 0 | **increase** | **increase** | 0 | 0 | 0 |
| Clearcut uniform | 0 | 0 | 0 | 0 | **increase** | **increase** | 0 | 0 | 0 |
| Thinned | 0 | 0 | 0 | 0 | 0 | 0 | 0 | 0 | 0 |
| Reference | 0 | 0 | 0 | 0 | 0 | 0 | 0 | **decrease** | 0 |
| **C. Pre-Post comparison of PLS Model predictions** | | | | | | | | | |
| Clearcut variable | **B/C*** | – | C/C | A/AB | **C/AB*** | **B/A*** | – | AB/AB | – |
| Clearcut uniform | AB/AB | – | A/A | AB/A | **C/A*** | **C/AB*** | – | A/AB | – |
| Thinned | AB/B | – | BC/BC | AB/AB | **C/B*** | BC/AB | – | A/AB | – |
| Reference | A/AB | – | B/AB | AB/AB | BC/BC | BC/ABC | – | AB/B | – |

An asterisk and bold text indicates significant differences between pre- and post-harvest periods. B). "0" indicates the pre- and post-harvest confidence intervals (CI) overlapped; "increase" indicates that the lower bound of the post-harvest CI was greater than the upper bound of the pre-harvest CI. "decrease" indicated pre-harvest CI was greater than post-harvest. C). Pre-post comparisons of significance for PLS model predictions for each attribute were based on 95% confidence intervals using the wild bootstrap. Within an attribute, pre/post with the same letters are not significantly different. "A" denotes the largest value, "B" is second largest, etc. within a response variable across groups. The lettering on PLS model predictions is based on comparing the Box-Cox model predictions, not transformed back the original scale. " –" indicates that the $R^2CV$ was negative and therefore the significance was not reported. Dissolved organic carbon (DOC), Soluble reactive phosphorus (SRP), Dissolved organic nitrogen (DON), Dissolved inorganic nitrogen (DIN), Chlorophyll a (Chl a).

**Table 3. Statistical summary from Partial Least Square models comparing pre- to post-harvest responses in headwater watersheds.**

| Treatment group | Watershed | Canopy closure | DOC | SRP | DON | DIN | DIN:SRP Molar Ratio | Epilithon biomass | Chl a |
|---|---|---|---|---|---|---|---|---|---|
| Clearcut variable | GS3 | **BC/D*** | CD/CD | G/G | ABC/ABC | **D/AB*** | **C/A*** | AB/AB | AB/ABC |
| Clearcut variable | UM2 | **AB/C*** | BC/BC | G/FG | ABC/ABC | **CD/A*** | **BC/A*** | AB/AB | BC/C |
| Clearcut variable | UM3 | **B/C*** | BC/BC | E/E | BC/BC | CD/BC | BC/BC | B/BC | B/BC |
| Clearcut uniform | PH1 | ABC/BC | AB/A | E/E | ABC/ABC | **CD/AB*** | C/BC | BC/CD | AB/ABC |
| Clearcut uniform | PH2 | AB/AB | BC/ABC | A/AB | BC/C | **CD/AB*** | C/C | BC/CD | ABC/BC |
| Clearcut uniform | PH4 | AB/AB | C/CD | DE/D | BC/C | **C/A*** | B/BC | AB/ABC | BC/BC |
| Thinned | GS2 | AB/ABC | BC/ABC | FG/FG | B/BC | CD/BC | BC/BC | BC/CD | AB/A |
| Reference | UM1 | ABC/ABC | **D/E*** | FG/FG | BC/ABC | D/D | C/C | **C/D*** | C/C |
| Reference | GS1 | AB/AB | AB/AB | FG/F | AB/A | A/A | AB/A | AB/AB | AB/AB |
| Reference | PH3 | A/A | CD/CD | B/B | ABC/BC | B/B | C/C | A/AB | AB/ABC |
| Reference | RK1 | BC/BC | CD/CD | C/C | ABC/ABC | D/D | C/C | BC/CD | BC/C |
| Reference | RK3 | AB/AB | B/AB | FG/FG | BC/C | BC/BC | BC/B | BC/C | BC/C |

Significance in Partial Least Squares (PLS) regression models was based on 95% confidence intervals calculated using Wild Bootstrap. Within a site, pre/post comparisons with the same letters are not significantly different. An asterisk and bold text indicates significant differences between pre- and post-harvest periods. "A" denotes the largest value, "B" is second largest, etc within a response variable among sites. The lettering on PLS model predictions is based on comparing the Box-Cox model predictions, not transformed back the original scale. Dissolved organic carbon (DOC), Soluble reactive phosphorus (SRP), Dissolved organic nitrogen (DON), Dissolved inorganic nitrogen (DIN), Chlorophyll a (Chl a).

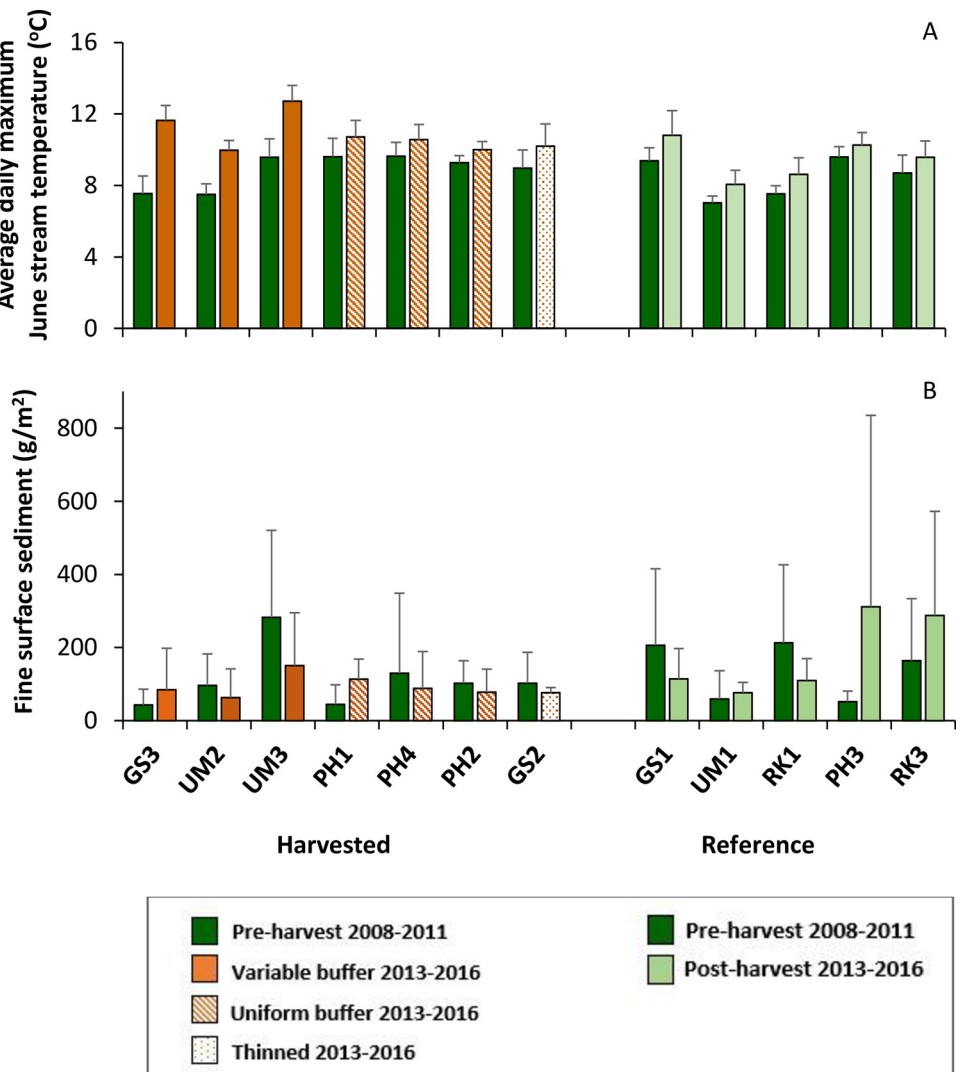

**Fig 3. Stream temperature and fine surface sediment.** Headwater sites are organized by harvest treatment and organized from left to right by the percentage of overstory light reaching the stream post-harvest. A) Average daily maximum stream temperatures in June during pre and post-harvest periods. B) Mass of fine surface sediment at annual sampling in June, for pre- and post-harvest periods. Bars show site averages and standard deviation.

In the headwater streams, substrates were primarily gravel to cobble with some areas of bedrock. The average mass of fine benthic sediment (Fig 3B) did not significantly change post-harvest in any of the treatment groups (Table 2), within individual headwater watersheds (Fig 3B, Table 3, S2 Table) or at downstream sites (S3 Table).

## Water chemistry

During our annual early summer sampling, concentrations of dissolved nutrients and DOC in these forested headwater streams were low and varied among watersheds. Pre- and post-harvest responses were not significantly different for headwater treatment groups for DOC and DON (Table 2) or for individual headwater watersheds (Table 3). Average DOC ranged from 0.240 to 0.883 mg C/L (Fig 4A). Because the $R^2$ CV was negative for DOC in the treatment group PLS model (S1 Table), the significance for the PLS model was not reported for DOC

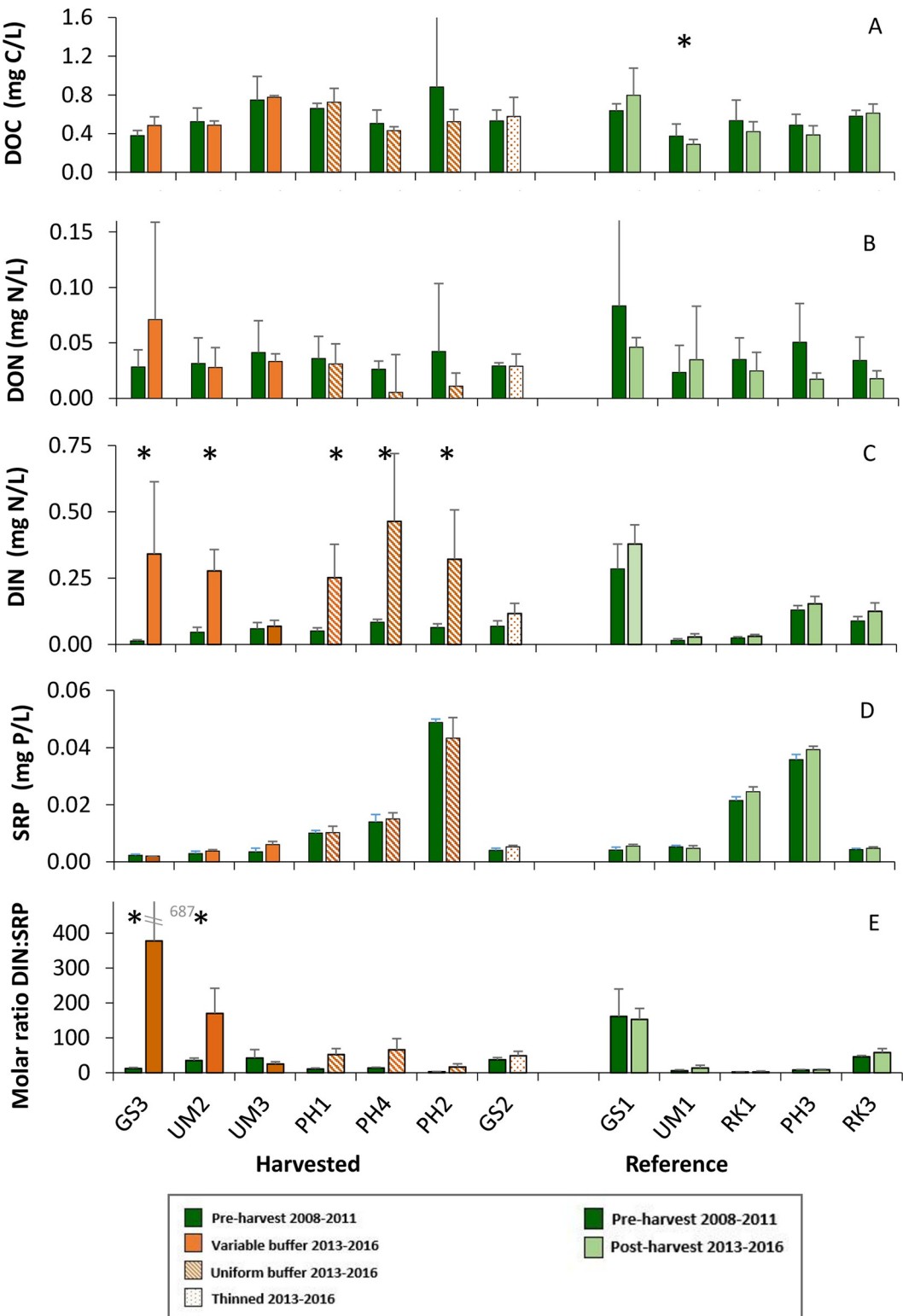

**Fig 4. Average concentrations of dissolved nutrients and carbon at headwater sites.** Sites are organized from left to right by percentage of overstory light reaching the stream post-harvest. A) Dissolved organic carbon (DOC), B) Dissolved organic nitrogen (DON). C) Dissolved inorganic nitrogen (DIN), D) Soluble reactive phosphorus (SRP), E) Molar ratios of DIN:SRP. Bars show averages plus standard deviation during early summer annual sampling. Asterisks indicate that watershed values were significantly different using PLS analyses (Table 3).

(Table 2C). Average DON varied from 0.005 to 0.083 mg N/L (Fig 4B). DIN showed significant increases post-harvest at all clearcut treatment groups (Table 2). DIN concentrations post-harvest in clear-cut treatments were approximately six times greater than pre-harvest, relative to a slight increase at the reference and thinned watersheds post-harvest (1.3x and 1.7x, respectively) (Fig 4C). The greatest change pre- to post-harvest in DIN occurred at GS3 where average concentration increased from 0.008 mg N/L pre-harvest to 0.334 mg N/L in the fourth year post-harvest (S4 Table). DIN was predominately composed of nitrate (S4 Table) and concentrations of nitrate in 2013, the first year post-harvest were not the highest observed in the post-harvest period. Nitrate concentrations peaked in 2015 at many sites. SRP concentrations showed very little change following harvest (Fig 4D, Table 2). Some watersheds, PH2, PH3, and RK1, had substantially higher SRP concentrations than others. Pre- to post-harvest changes in DIN:SRP molar ratio mirrored the increases in DIN for watersheds clearcut with variable buffers, but showed much smaller increases post-harvest in the clearcut with uniform buffer group (Fig 4E). Molar ratios for both clearcut treatment groups were significantly greater post-harvest (Table 2), but in the analysis for individual watersheds, only GS3 and UM2 were significantly higher (Table 3). The average molar ratios of DIN:SRP concentrations in the headwaters ranged from 2.5 to 161 pre-harvest and up to 377 post-harvest at GS3 (Fig 4E).

The PLS model for individual watersheds showed significant post-harvest increases in DIN in four of the six clearcut watersheds (Table 3). The thinned watershed, GS2, showed a significant increase in DIN post-harvest in the PLS model of treatment groups but not in the watershed PLS model. Because the watershed PLS model had a higher $R^2CV$ than the associated model of treatment groups, the watershed model could be viewed as more reliable.

Nutrient and DOC concentrations at downstream sites were very similar to those in the headwaters (Fig 5). As observed at headwater sites, downstream DOC, SRP, and DON did not show significant differences between pre- and post-harvest periods (S3 Table). The increased concentration of DIN post-harvest in clearcut headwaters did not result in a consistent significant increase of DIN at their downstream study sites; the analysis of overlapping 95% confidence intervals showed DIN to be significantly greater post-harvest period downstream of harvested headwaters, but the other analyses did not (S3 Table). Interestingly, the downstream of reference site showed significant differences for DIN and DIN:SRP molar ratios post-harvest. The comparison of 95% confidence intervals and the PLS Models indicated that the DIN concentrations decreased post-harvest at the downstream reference site, possibly related to dryer and warmer conditions. Comparisons from the PLS model were not reported for DOC, DON, or molar ratio because the $R^2CVs$ were negative, indicating that the model quality was poor (S3 Table).

## Epilithic algal biomass and chlorophyll *a*

Average epilithon biomass at headwater sites ranged from 3.8–14.3 g AFDM/m$^2$ (Fig 6A). Post-harvest, epilithon biomass was not significantly different from pre-harvest levels in any of the harvested headwater groups (Table 2) or individual headwater watersheds that were harvested (Table 3). A reference watershed (UM1) had significantly lower biomass post-harvest (Table 3) and likely influenced KS2 test that showed significantly different distribution for the reference group (Table 2). In the PLS watershed model, the multi-comparison lettering (Table 3) based on wild bootstrap prediction confidence intervals, was indicative of a general decrease in epilithon during the post-harvest period, in agreement with Fig 6. The $R^2$ CV was zero for epilithon in the PLS model for headwater treatments (S1 Table), therefore the model significance was not reported (Table 2C).

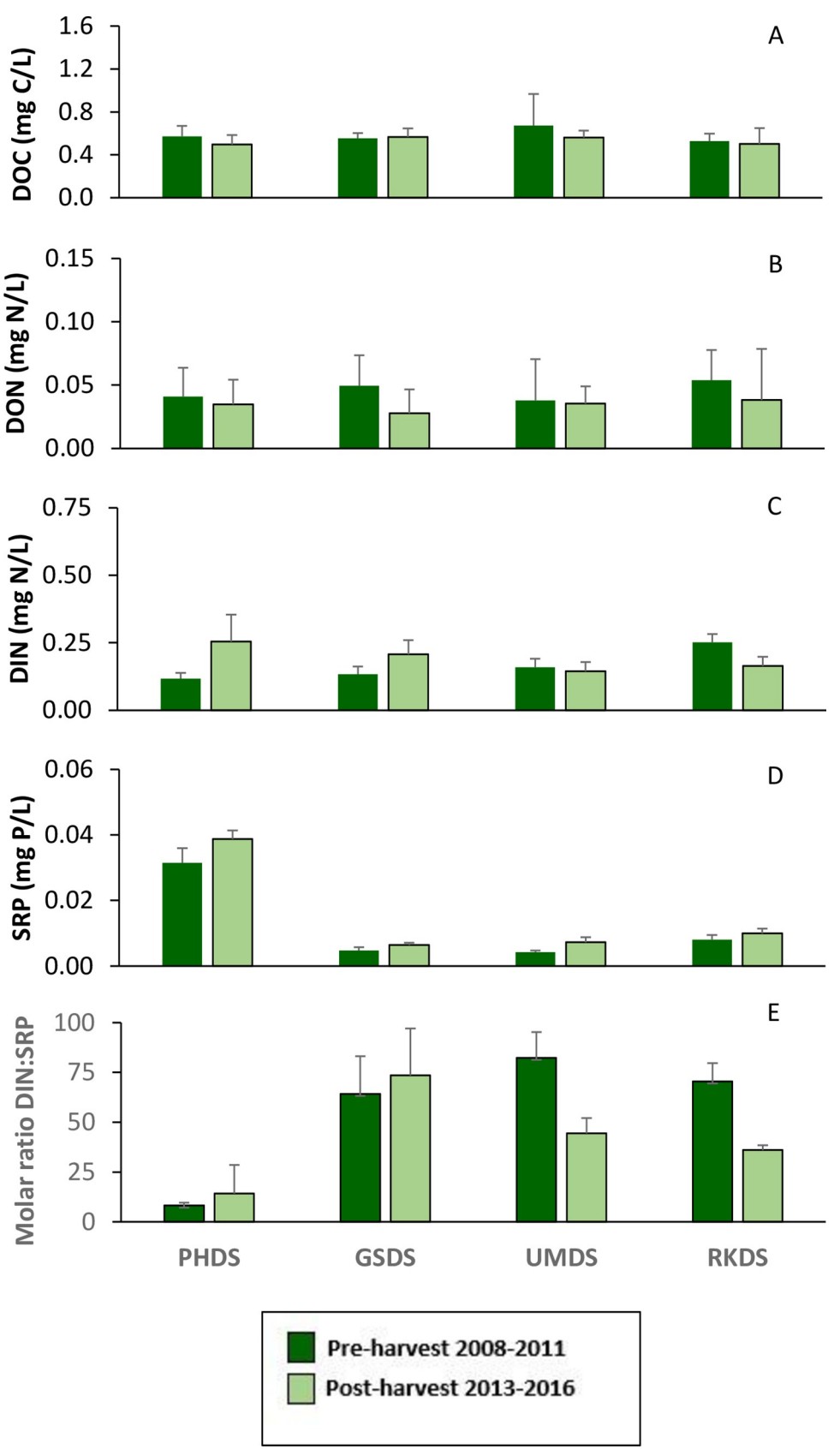

**Fig 5. Average concentrations of nutrients at downstream sites during pre- and post-harvest periods.** A) Dissolved organic carbon (DOC), B) Dissolved organic nitrogen (DON), C) Dissolved inorganic nitrogen (DIN), D) Soluble reactive phosphorus (SRP), E) Molar ratio DIN:SRP. Bars show averages plus standard deviation during early summer annual sampling. RKDS is the reference for the downstream sites. UMDS is downstream of variable buffer sites, PHDS is downstream of uniform buffer sites and GSDS has mixture of harvest groups upstream.

Average chlorophyll *a* concentration ranged from 3.3–13.0 mg/m² (Fig 6B). There were no significant differences in chlorophyll *a* pre- to post-harvest in clearcut with variable buffer group or in the thinned treatment (Table 2). The KS2 analyses, which compares distributions

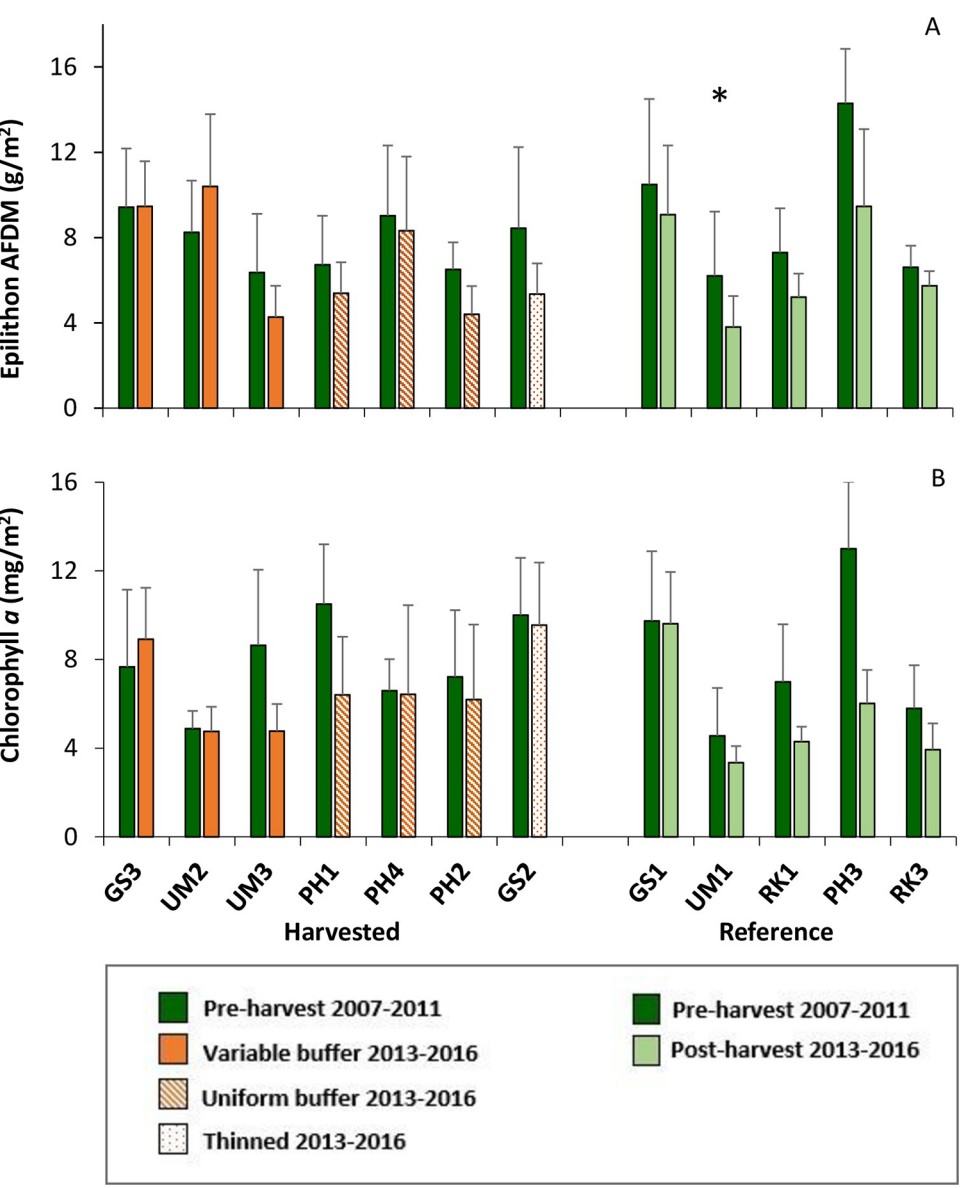

**Fig 6. Average epilithon biomass and chlorophyll *a* concentration at headwater sites.** Sites are organized from left to right by percentage of overstory light reaching the stream post-harvest. A) Average epilithon biomass as ash free dry mass (AFDM) plus standard deviation; B) Average chlorophyll *a* concentration plus standard deviation. Samples were collected annually in early summer. Asterisks indicate which watersheds showed significant differences between pre- and post-harvest periods using PLS analyses (Table 3).

**Table 4. The most common and second most common algal taxa in selected headwater streams, based on cell counts of epilithon samples.**

| Treatment group | Watershed | Period | Most Common Algal Taxon [1] | Second Most Common Algal Taxon [1] |
|---|---|---|---|---|
| Clearcut variable | GS3 | Pre | *Gomphonema minutum* (H) | *Meridion circulare* (L) |
| | | Post | *Achnanthidium minutissimum* (L) | – |
| Clearcut variable | UM2 | Pre | *Planothidium lanceolatum* (L) | *Achnanthidium minutissimum* (L) |
| | | Post | *Achnanthidium minutissimum* (L) | *Achnanthidium cf. linearis* (L) |
| Clearcut variable | UM3 | Pre | *Achnanthidium minutissimum* (L) | *Achnanthidium cf. deflexum* (L) |
| | | Post | *Achnanthidium minutissimum* (L) | *Achnanthidium cf. linearis* (L) |
| Clearcut uniform | PH4 | Pre | *Rhoicosphenia abbreviatum* (L) | *Nitzschia inconspicua* (L) |
| | | Post | *Rhoicosphenia abbreviatum* (L) | *Planothidium lanceolatum* (L) |
| Reference | UM1 | Pre | *Gomphonema minutum* (H) | *Planothidium lanceolatum* (L) |
| | | Post | *Planothidium lanceolatum* (L) | *Gomphonema minutum* (H) |

[1] Growth forms are indicated as low (L) and high (H) profiles, based on classification by Rimet and Bouchez, 2012 [81].

but not means, was the only test that suggested significant differences for post-harvest chlorophyll *a* in the clearcut with uniform buffer treatment (Table 2A). The Reference group showed a significant decrease in the comparison of 95% BCa confidence internals (Table 2B). Although PLS analysis of Treatment groups showed no significant change for any group (Table 2C), the PLS model $R^2CV$ for chlorophyll *a* was low, suggesting weak predictive power (S1 Table). No significant differences were identified for individual headwater watersheds (Table 3).

An analysis of the algal cell counts from a subset of headwater sites found that the June algal community was primarily diatoms in pre-harvest and post-harvest samples (Table 4); no filamentous algae were present. *Achnanthidium minutissimum* was the most common taxon in the epilithic algal samples at three of the five sites where taxa were identified pre- and post-harvest. At four of the five sites, one of the two most common species was the same in pre- and post-harvest periods. No clear shifts in growth form were noted among the treatments.

At the downstream sites, epilithon biomass and chlorophyll *a* concentration in the downstream of harvested group were generally lower in the post-harvest period (Fig 7). The group of sites 'downstream of harvested" showed significantly different distributions for both epilithon mass and chlorophyll *a* concentration between pre- and post-harvest periods using the KS2 test (S3 Table) and epilithon biomass significantly decreased in the analysis of 95% confidence intervals. Using the PLS model of pre- to post-harvest comparisons, neither epilithon nor chlorophyll *a* showed significant differences across these time periods.

Chlorophyll *a* concentrations were significantly correlated with epilithon biomass during pre-harvest period ($R^2 = 0.446$, Fig 8A; $R^2 = 0.521$, Fig 8B). However, during the post-harvest period, the correlations were significant only for reference sites ($R^2 = 0.444$, Fig 8B). Chlorophyll *a* was not significantly correlated with canopy closure at reference or harvested sites, pre- or post-harvest (Fig 8C and 8D). Epilithon was not correlated with DIN at harvested sites (Fig 8E); only at reference sites during post-harvest period was the correlation between epilithon and DIN significant ($R^2 = 0.300$, Fig 8F). The correlation between epilithon and DIN: SRP molar ratios was significant for harvested sites in the post-harvest period ($R^2 = 0.149$, Fig 8G) and likely greatly influenced by several high molar ratios. No correlations for epilithon and ratios at reference sites (Fig 8H).

## Discussion

Retention of vegetated riparian buffers can reduce impacts of forest harvest on stream ecosystems. In the Trask River Watershed study, all streams in harvested forests showed increased

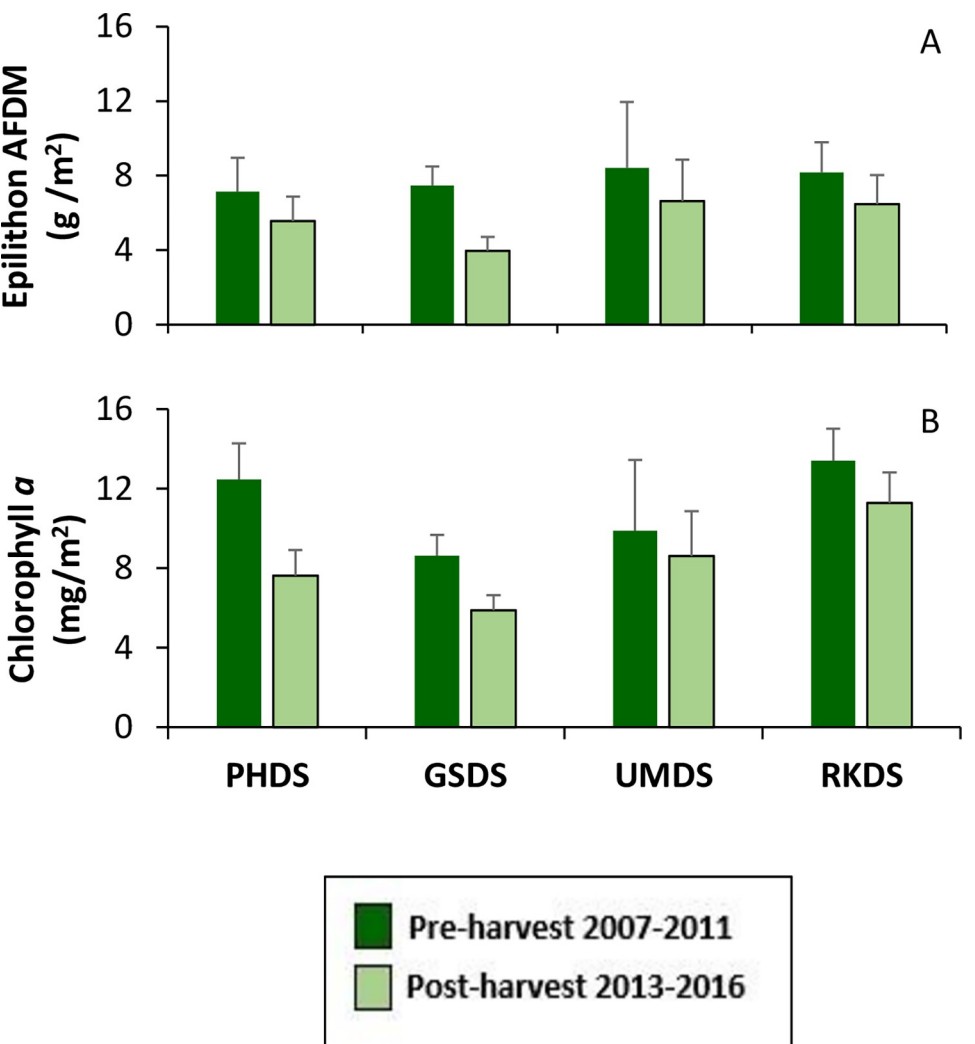

**Fig 7. Average epilithon biomass and concentration of chlorophyll a at downstream sites during pre- and post-harvest periods.** A) Epilithon biomass as ash free dry mass (AFDM); B) Chlorophyll *a* concentration. Bars show early summer average plus standard deviation. RKDS was the reference for the downstream sites. UMDS was downstream of variable buffer sites, PHDS was downstream of uniform buffer sites and GSDS had mixture of harvest groups upstream.

DIN regardless of the width of riparian buffers, and no change in concentrations of dissolved organic carbon, soluble reactive phosphorus, or mass of benthic sediment. Canopy closure remained similar to the pre-harvest period at the sites with uniform 12 m buffers but dramatically increased at the sites with variable riparian buffers. Despite increased levels of both light and dissolved inorganic nitrogen in clearcut watersheds with variable riparian buffers, we did not observe increased standing stocks of epilithon or higher concentrations of chlorophyll *a*, in contrast to the common expectation of increased autotrophic resources in response to increased light and nutrients. Headwater streams have been overlooked in many prior regulatory discussions; therefore these findings are valuable in expanding our understanding of what conditions and factors are major drivers of responses in small headwater streams. Though our use of multiple analyses, we have greater confidence in the statistical significance of these relevant management practices.

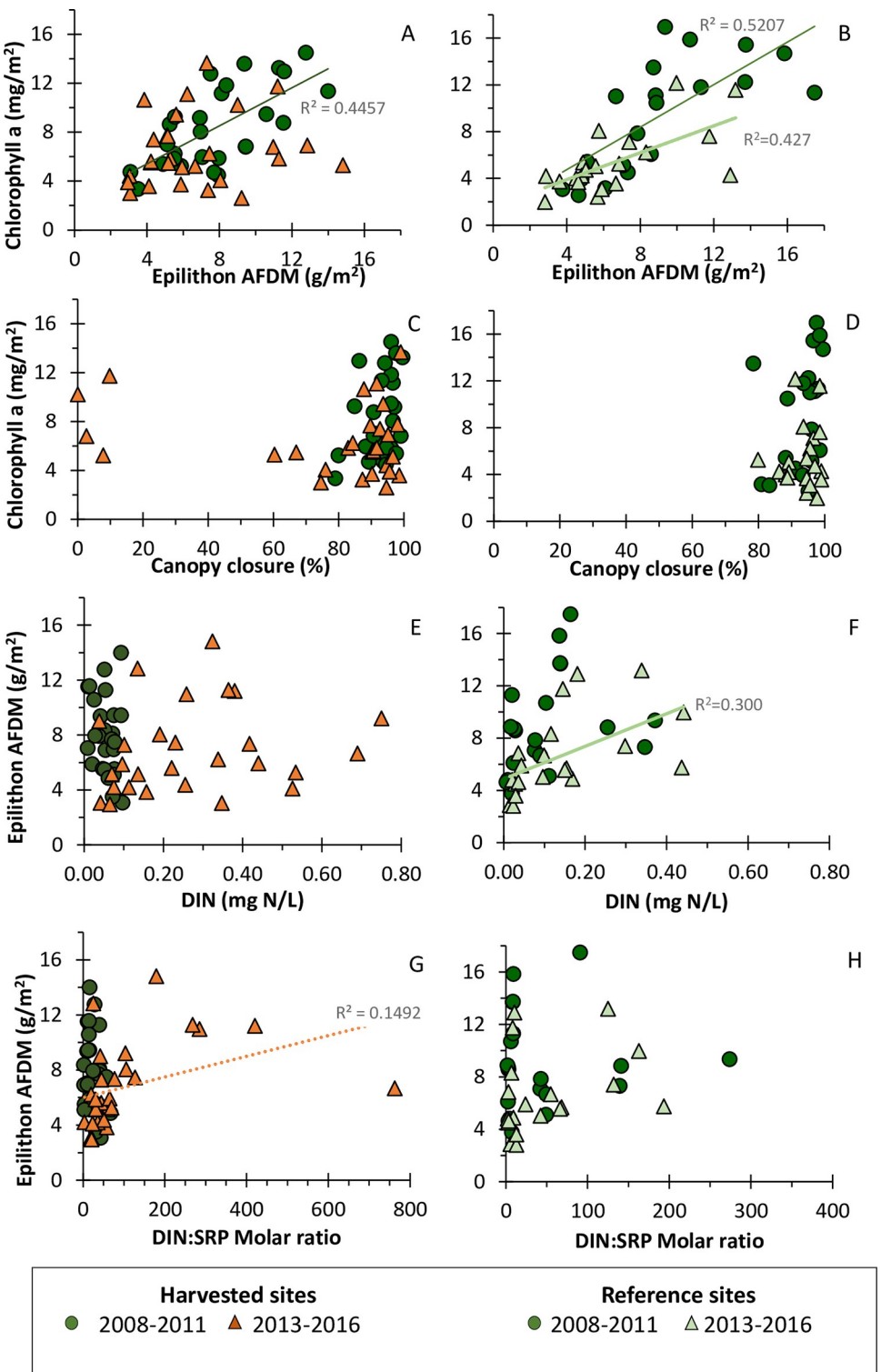

**Fig 8. Correlations between chlorophyll *a*, epilithon and instream factors at headwater sites.** These correlations compare site averages for each year and site during pre- and post-harvest periods. Headwater sites that were harvested are in the left column (A, C, E, G) and those were reference sites are in the right column (B, D, F, H). Chlorophyll *a* versus epilithon (A, B); Chlorophyll *a* versus canopy closure (C, D); Epilithon versus DIN (E, F); Epilithon versus molar ratios of DIN:SRP (Dissolved Inorganic Nitrogen: Soluble Reactive Phosphorus) (G, H). Data from pre-harvest period are indicated as dark green circles and from the post-harvest period are shown as triangles. $R^2$ values are included for the correlations where p-values are equal to or less than 0.05.

## The challenges of distinguishing statistical significance in landscape experiments

There are multiple challenges to distinguishing statistical significance in field experiments. Research into responses to landscape treatments are conducted over a large area and for a number of years. These large-scale field experiments are expensive and labor-intensive and consequently have low sample sizes [37, 39, 93], and the focal metrics often exhibit high spatial and temporal variability [54]. Our study design planned that individual watersheds would be replicates of a specific harvest treatment, but study watersheds are never true replicates [34]. Our study sites within each treatment group showed variability in site conditions and responses [19, 68]; and in this study, we explored this variability by including examination of pre- and post-harvest responses within each watershed. In addition to treatments, we included reference watersheds to control for the influence of variable climatic conditions during the 10 years of the study. The analyses of treatment groups were crucial for evaluation of forestry best management practices, while the analyses of individual watersheds addressed the variability in responses across sites.

Our results from this complex landscape-scale experiment have potentially broad implications for forest management and policy issues. We recognize that statistical significance becomes a major factor for the incorporation of research findings into policy and management; To address the need of statistical robustness, we used multiple statistical approaches to assess pre-and post-harvest differences with the belief that consensus among methods would provide stronger evidence than any one method by itself. The use of multiple methods was further warranted given the data challenges of spatial and temporal dependency, small sample sizes, and the potential of not finding an acceptable PLS model given the stringent quality criterion of $R^2CV$. Multiple statistical tests allowed us to evaluate means (BCa bootstrap) and distributions (KS2) for pre-post comparisons, and simultaneously assess responses across all treatments or watersheds using PLS models. The non-parametric KS2 and BCa bootstrap both used small sample sizes which can be problematic [94–96], but by bootstrapping, the confidence intervals were considered conservative (i.e. larger) for non-independent data [97]. Our multi-analysis approach leveraged the study design of pre-post, reference-treatment data, but unlike most Before-After-Control-Intervention (BACI) and repeated measures tests; our analyses of significance did not depend upon normally distributed data for comparisons.

## Stream chemistry responses to riparian and watershed harvest

The increase in DIN concentrations concur with results from other studies of stream chemistry responses to forest management [35, 37, 98, 99]. Early studies in non-nitrogen limited regions found extremely high concentrations of nitrate following harvest [37, 42, 100], especially during storm flows. Our sampling timing was focused on early summer, non-storm periods, which in these oligotrophic ecosystems, is a time when nutrients are generally at their minimum concentrations [101, 102]. The magnitude of DIN increase in this study was substantial relative to the pre-harvest period, but the absolute concentrations were relatively low and comparable to studies in western US that documented small increases in nitrogen after forest harvest without riparian buffers [35, 39, 93].

Significant increases in DIN concentrations occurred even where riparian buffers were retained. Maintaining forested riparian zones during watershed harvest is a widely recognized best management practice for protecting water quality by reducing nutrient and sediment delivery to streams [9, 16]. Stream nitrogen increases following removal of vegetation have been hypothesized to be driven by a reduced demand for nitrate where there is less vegetation, coupled with mobilization of nitrate related to soil disturbance, decaying organic matter and

microbial nitrification [38, 99]. The effectiveness of riparian buffers to intercept, retain and process nutrients has been related to the width of the buffer, with the recognition that multiple other landscape factors, including slope, soil type, and depth of groundwater also provide key functions [16, 23, 43]. Our high-gradient watersheds have a mix of soil types [64] and a legacy of soil disturbances related to historical wildfire and subsequent salvage logging [19, 70] that may have altered connections between subsurface hillslope flow paths and rooting zones of riparian vegetation [16]. Although transport of inorganic nitrogen to the Trask streams may not have been fully intercepted in watersheds with wider buffers, the wider buffers provided other water quality functions, including shade that prevented stream temperature increases [19].

Responses of carbon and nutrients other than DIN to forest harvest and removal of riparian buffers have been mixed. Similar to findings from other studies in the Pacific Northwest [42, 100, 104], we did not see increased concentrations of DON, DOC, or phosphorus. However, in other regions such as boreal forests, researchers have observed increased concentrations and export of DON and DOC following harvest [40, 101]. Although increases of stream phosphorus following harvest are not common [42], adsorption of P to soil and DOC can enhance its mobility and export [101], with site specific differences in soil and vegetation type likely controlling these dynamics. Even where phosphorus occurs naturally at high concentrations, including western Cascade Mountains [42, 106], an increase in phosphorus concentrations has not generally been observed following forest harvest or watershed disturbances [35, 101, 107], unless substantial transport of sediment occurred [35, 39].

Downstream propagation of responses to headwater disturbances can be a concern [11] and has been observed for water quality [103, 104], especially during stormflows [39]. In nitrogen-limited ecosystems, nitrogen spiraling, uptake, and transformation would be expected to reduce instream concentrations of N as the water flows downstream [105, 106]. At the Trask sites downstream of harvest, downstream propagation of upstream DIN increases was not definitive; our statistics showed mixed results, albeit with small sample sizes.

## Lack of epilithic algal responses

Increased nutrients [45, 78, 107, 108] as well as increased light [29, 109] have long been recognized to stimulate autotrophic responses in aquatic systems. When increased nutrients coincide with increased light, many studies have documented an increase in autochthonous resources and productivity instream [21, 29, 48, 55]. Therefore, at our sites where both DIN and light increased post-harvest, we expected but did not observe increased standing stocks of epilithon and/or changes in chlorophyll *a*. We surmised that lack of response might be related to limitation of individual nutrients and/or co-limitation of nitrogen and phosphorus [58, 59, 107]. Although the Trask DIN concentrations increased after harvest and approached proposed "breakpoint" and saturation concentrations associated with high chlorophyll values [110], phosphorus concentrations in the Trask (except in PH2, PH3, and RK1) were generally lower than those noted by other studies and used for predicting algal standing stocks and chlorophyll *a* [110, 111]. Historically, stream water N:P ratios (evaluated here as DIN:SRP) have been proposed as predictors of algal responses and have been included regularly in comparisons [55, 110, 112], while others have suggested that they might not be strong indicators of nutrient limitation [108]. Across our wide range of N:P ratios across sites, some of which were in optimum ranges for algal growth, epilithic algal stocks did not increase following harvest. Our correlations between algae responses and potential drivers of increased algae showed few relationships (Fig 8). In very small streams, co-limitation may involve multiple mechanisms including nutrients [58, 107] in combination with light [113] plus other factors.

The taxonomic composition of algal assemblages can influence their responses to disturbance [108]. Multiple studies of responses to forest harvest have observed taxonomic shifts and increased filamentous green algae following the opening of the riparian canopy [2, 45, 46] and deposition of benthic sediment [56], yet the Trask algal assemblages were consistently dominated by small, low growing diatoms during pre-and post-harvest periods. *Achnanthidium minutissimum* is considered an early successional species, that competes well as in shaded headwater streams with low nutrients and high water velocities [79, 114] but can be replaced by other taxa when inorganic nutrients and light increase [115]. Yet our small study streams at the upper tips of the stream network had minimal green algal taxa to take advantage of post-harvest changes. Diatoms are frequently used as indicators of water quality and watershed condition [79, 114, 116]; the diatoms in the Trask streams were indicators of high water quality and cold waters. Some diatom taxa are more responsive than others to changes in nutrients; Stelzer and Lamberti [112] noted that *Achnanthidium minutissimum* showed sensitivity to N:P ratios while other diatom taxa, such as *Gomphonema* sp., were sensitive to only increases in nitrate. Increased light availability can lead to shifts in community composition from low, prostrate to high upright algal growth forms (e.g., Bixby et al. [117]), yet the Trask algal assemblages showed no shifts in growth forms associated with increased light availability. Light flecks [57] in shaded stream beds, including reference sites, may have supported microhabitats of algal assemblages with higher profile taxa (e.g., *Gomphonema minutum*).

We wondered if the lack of observed responses in epilithon could be due to the timing of our samplings (i.e., early summer), as seasonality can be a major factor for composition of diatom assemblages as well as green algae and summer is the time of highest activity and biomass [59, 108]. To assess if the timing of our sampling affected the results about epilithic responses, we examined the algal standing stocks and chlorophyll *a* concentrations later in the summer when temperatures were warmer and flows lower. However, additional sampling in July showed generally lower chlorophyll *a* and similar biomass of epilithon as in June (S2 File), thereby confirming that we were not missing substantial algal responses post-harvest.

A final consideration of influences of autochthonous response to disturbances involves the potential influences of instream grazers [55, 109, 115]. The Trask stream food webs communities had generally low abundances of grazers preharvest. We documented that macroinvertebrate scrapers and grazers did not increase post-harvest [68], which was expected if more autochthonous resources had become available. Tailed frog tadpoles were present in low densities in four of our treated headwaters sites (UM2, PH1, PH2, GS2) and in three of our reference sites (UM1, RK1, RK3) [72]. Although other studies have noted the role of grazers in algal responses post-treatment [21, 109], in the Trask watersheds, the presence of tailed frogs did not help explain differences in chlorophyll *a* or standing stocks of algae pre-post treatment or among sites.

Harpole et al. [58] noted that minimal or null responses of autotrophic community to increased light and nutrients "are rarely discussed in the nutrient limitation literature". In our review of the primary literature related to algal responses to disturbances, we found only few studies where increased nitrogen or combination of light and nitrogen did not lead to increased algal standing stocks and chlorophyll *a* [59, 118, 119]. The lack of studies showing null responses to increased light and nutrients might be partially attributed to a bias in the published literature towards studies that show significant responses after disturbances or treatments [54, 55, 107, 120]. The few published studies where null responses to increased nutrients and light were documented have suggested the lack of observed response was due to low phosphorus [118] or seasonally shifting N:P ratios [108, 121]. Frequently, natural resource managers and some researchers have expectations that removal of forest cover or increases in nutrients will result in straightforward and predictable algal increases that transfer to upper

trophic levels [55] for more productive fisheries [49, 51–53, 122], based on the general paradigm that light and nutrients stimulate instream primary producers. Meta-analyses [54, 55] have suggested responses to nutrient enrichment are contingent on background concentrations of inorganic N and P with interactions among nutrients, light and turbidity.

## Conclusions and management implications

The evolution of forest harvest methods and improvements in Best Management Practices have resulted in smaller magnitude of post-harvest impacts than historically observed [9, 18, 19, 100]. However, the stakes to quantify the impacts of forest management accurately and to inform best management practices and policies continue to be high and regularly associated with controversy [32, 61, 123], especially in landscapes with mixed ownerships and management objectives. Our inclusion of multiple analytical tests allowed us to examine the variability in responses and the concurrence of statistical responses. This provided more certainty around our findings and better information for managers and policy makers in their planning and evaluations of responses to management.

Disturbance from forest harvest in these small watersheds led to increased instream dissolved nitrogen broadly while the wider riparian buffers maintained lower light levels representative of forested headwater streams. However, this study reminds us that not all streams respond similarly to forest harvest and removal of riparian vegetation; our results did not fit the general expectation that increased light and or increased stream nitrogen after watershed disturbances stimulate algal responses. We postulate that multiple factors, including co-limitation of nutrients, driven by low phosphorus concentrations, which did not change, plus an algal community dominated by low light adapted diatoms rather than green algae, contributed to our findings of no change in biomass of epilithon or concentrations of chlorophyll *a*. Management and restoration activities which aim to increase abundances and standing stocks of fish by opening riparian canopies and adding nutrients would benefit from prior evaluation of algal community composition and co-limitation of nutrients before undertaking major modifications.

## Supporting information

**S1 Table. Details of PLS, Wu jacknife, Wild Bootstrap models for headwater treatment groups.** Box-Cox transformations were used with the goal of improving model quality (See S1 File for details on Box-Cox transformations). An exponent of 1 indicates that raw data were used. For normality tests, bold indicates that the null hypothesis of normal residuals was rejected. For Levene's test, bold indicates that the null hypothesis of a common variance across groups of a given factor was rejected. Significance was not reported in Table 2 when $R^2CV$ was negative for PLS. For Wu Jackknife and Wild Bootstrap, bold indicates regression coefficients were significantly different than zero. We did not assume the regressor rows were independent and consequently used the Wild Bootstrap and Wu Jackknife for discerning model parameter significance. Interaction predictions are PLS model predictions transformed back to the original scale of the data. Dissolved organic carbon (DOC), Soluble reactive phosphorus (SRP), Dissolved organic nitrogen (DON), Dissolved inorganic nitrogen (DIN), Chlorophyll a (Chl a). (DOCX)

**S2 Table. Details of PLS, Wu jackknife and Wild Bootstrap models for headwater watersheds.** Box-Cox transformations were used with the goal of improving model quality (See S1 File for details on Box-Cox transformations). An exponent of 1 indicates that raw data were used. For normality tests, bold indicates that the null hypothesis of normal residuals was

rejected. For Levene's test, bold indicates that the null hypothesis of a common variance across groups of a given factor was rejected. For Wu Jackknife and Wild Bootstrap, bold indicates regression coefficients were significantly different than zero. We did not assume the regressor rows were independent and consequently used the Wild Bootstrap and Wu Jackknife for discerning model parameter significance. Interaction predictions are PLS model predictions transformed back to the original scale of the data. Dissolved organic carbon (DOC), Soluble reactive phosphorus (SRP), Dissolved organic nitrogen (DON), Dissolved inorganic nitrogen (DIN), Chlorophyll a (Chl a).
(DOCX)

**S3 Table. Statistical summaries and details of PLS, Wu jackknife and Wild Bootstrap models for downstream sites. A)** Kolmogorov-Smirnov two sample test p-values compare post- and pre- empirical cumulative distribution functions for downstream site groups (downstream of harvest and downstream of reference). **B)** Comparison of 95% Bias Corrected accelerated (BCa) Bootstrap confidence intervals (CI) for observed pre- and post-harvest treatment means for each response variable. "0" indicates the pre- and post-harvest CI overlapped; "Increase" indicates that the lower bound of the post-harvest CI was greater than the upper bound of the pre-harvest CI. "decrease" indicated where pre-harvest was greater than post-harvest. **C)** Pre/Post comparisons of significance for PLS model predictions for each attribute, based on 95% Wild Bootstrap confidence intervals. When $R^2CV$ was negative, significance was not reported, indicated by "- -". Within a group, pre/post with the same letters are not significantly different: "A" denotes the largest value, "B" is second largest, etc, for that attribute. Box-Cox transformations were used with the goal of improving model quality. For Wu Jackknife and Wild Bootstrap, bold indicates regression coefficients were significantly different than zero. We did not assume the regressor rows were independent and consequently used the Wild Bootstrap and Wu Jackknife for discerning model parameter significance. Dissolved organic carbon (DOC), Soluble reactive phosphorus (SRP), Dissolved organic nitrogen (DON), Dissolved inorganic nitrogen (DIN), Chlorophyll a (Chl a).
(DOCX)

**S4 Table.** Concentrations of A) stream nitrate ($NO^3$) and B) ammonium ($NH_4$) at each site, each year at the time of early summer sampling. "–" indicates no sample collected. Additional Trask River Watershed study chemistry data are available through US Forest Service Research Data Archive: https://doi.org/10.2737/RDS-2022-0002.
(DOCX)

**S1 File. Details of Box-Cox transformations.**
(DOCX)

**S2 File. Epilithon biomass and chlorophyll *a* concentrations in June and July 2016.**
(DOCX)

## Acknowledgments

Bob Bilby, Liz Dent, Maryanne Reiter, Jason Dunham, Judy Li, Janel Sobota, Arne Skaugsat, and Bill Gerth were invaluable research partners. We thank Rich Van Driesche, Kylie Meyer, Brent Morrissette, Tim Glidden, Jenn King, Mark Meleason, and Katie Whitehead for multi-year field assistance plus many others too numerous to list. We appreciate the analyses, field maps and Fig 1 by David Hockman-Wert. Any use of trade, firm, or product names is for descriptive purposes only and does not imply endorsement by the U.S. Government.

## Author Contributions

**Conceptualization:** Sherri L. Johnson, Linda R. Ashkenas.

**Data curation:** Sherri L. Johnson, Linda R. Ashkenas.

**Formal analysis:** David C. Plaehn.

**Funding acquisition:** Sherri L. Johnson, Linda R. Ashkenas.

**Investigation:** Sherri L. Johnson, Alba Argerich, Linda R. Ashkenas, Rebecca J. Bixby.

**Methodology:** Sherri L. Johnson, Alba Argerich, Linda R. Ashkenas, Rebecca J. Bixby.

**Project administration:** Sherri L. Johnson, Linda R. Ashkenas.

**Resources:** Sherri L. Johnson.

**Software:** David C. Plaehn.

**Supervision:** Sherri L. Johnson, Linda R. Ashkenas.

**Validation:** Alba Argerich.

**Visualization:** Sherri L. Johnson, Linda R. Ashkenas, David C. Plaehn.

**Writing – original draft:** Sherri L. Johnson, Alba Argerich, Linda R. Ashkenas, Rebecca J. Bixby, David C. Plaehn.

**Writing – review & editing:** Sherri L. Johnson, Alba Argerich, Linda R. Ashkenas, Rebecca J. Bixby, David C. Plaehn.

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
