## [Decision Letter · Decision Letter 0]

28 Oct 2022

PONE-D-22-25727The paradox of stream nitrate enrichment and increased light yet no algal response following experimental manipulation of riparian zones and forested watershedsPLOS ONE

Dear Dr. Johnson,

Thank you for submitting your manuscript to PLOS ONE. After careful consideration, we feel that it has merit but does not fully meet PLOS ONE’s publication criteria as it currently stands. Therefore, we invite you to submit a revised version of the manuscript that addresses the points raised during the review process.I encourage to carefully consider all the comments of the reviewers, specially those from reviewers 2 and 3.

We look forward to receiving your revised manuscript.

Kind regards,

Clara Mendoza-Lera

Academic Editor

PLOS ONE

Journal Requirements:

a. You may seek permission from the original copyright holder of Figure 1 [#] to publish the content specifically under the CC BY 4.0 license.  

Reviewers' comments:

Reviewer's Responses to Questions

**Comments to the Author**

1. Is the manuscript technically sound, and do the data support the conclusions?

Reviewer #1: Yes

Reviewer #2: No

Reviewer #3: Partly

2. Has the statistical analysis been performed appropriately and rigorously? 

Reviewer #1: I Don't Know

Reviewer #2: No

Reviewer #3: I Don't Know

3. Have the authors made all data underlying the findings in their manuscript fully available?

Reviewer #1: Yes

Reviewer #2: No

Reviewer #3: Yes

4. Is the manuscript presented in an intelligible fashion and written in standard English?

Reviewer #1: Yes

Reviewer #2: Yes

Reviewer #3: Yes

5. Review Comments to the Author

Reviewer #1: This paper tests the response of algal biomass to different forest harvest, and riparian buffer practices. This study was conducted over ten-years and twelve streams, which is quite impressive and unusual. This work is interesting and high quality. While the work underling this manuscript is of good quality I have some questions about the statistics used to reach the conclusions of this work. Additionally, I think this manuscript could be more useful to other scientists and managers if it was shortened and simplified significantly. Please see my detailed comments below.

Major issues:

• Lines 257-263 and subsequent sections of results and discussion: This section describes the methods used to measure extra samples in July of 2016 and compare the measurements of algae on rocks to algae on tiles. Because the goals of this are not mentioned in the introduction (e.g., you do not have hypotheses stated in the introduction) this feels out of place here – particularly the comparison of between rocks and tiles. From the discussion section I understand that part of the motivation for this was to understand whether you were sampling too early in the season to see an effect of the treatments – but I don’t understand how the tiles fit into the goals of the present study. Additionally, the results from this don’t seem to really shed light on the findings of the main part of the study. I suggest you move all of this material to an appendix, and briefly reference the findings in the discussion.

• Statistical methods: This paper uses multiple statistical methods to test the same questions, about how the forest harvest/ buffer treatments effects in-stream variables. The inclusion of multiple statistical methods add considerable length to the manuscript, and complexity to an experimental design that is quite simple. This manuscript would be much clearer if there was a single method used – and less statistical tests generally.

• Line 307: Your hypotheses are all related to the effect of forest treatment across the watersheds which are essentially replicates. I don’t understand the need to look at changes in each watershed individually as you describe here.

• Line 437-439: Describes correlations between chlorophyll-a and light, I don’t see where these regressions are described in the methods.

• Figures: The goal of this manuscript is to investigate the instream responses to forest and riparian zone management. In the context of this, each stream is essentially a replicate – and I think it would make more sense to present them this way in the figures. Many of the figures show multiple bars for each of the streams. Presenting bar graphs grouped by treatments would fit better with your experimental design and would give the readers much less extraneous information. The figures in the present version could be moved to an appendix if you would like to keep that information available to a reader.

• Line 312-325. You describe the use of Wu’s jackknife to account for non-independence of measurements. I am more familiar with non-independence being addressed with random effects in mixed-effects models (e.g., as described in the books by Zuur 2009, or Wu 2009), than through bootstrapping. Reading Wu 1986 they seems to specifically caution against using their method in situations with non-independence. From Wu; “Our analysis of the bootstrap …. encounters similar difficulties in handling complex survey samples, where neither the independence nor the exchangeability assumption is satisfied (Rao and Wu (1985a)).” From my own exploration of your data it does appear that there is non-independence within dates and sites, with roughly half of the variation in the chlorophyll-a data explained by random effects for date and site. Please clarify if, and how the bootstrapping addresses these forms of non-independence in your data.

• Lines 484-496: This whole paragraph could be cut. I suspect any reader of this paper will understand that watersheds are not true replicates, and that environmental conditions are variable.

• Line 581-589: This paragraph seems very tangential to the stated goals of this paper and could be cut entirely or moved to an appendix.

• Line 612: The subheading for this section is “conclusions and management implications”. The management implications are not clear in the text of this section, or elsewhere in the paper. Please clarify what a manager should do with the information presented in this study.

• Line 617-620: I am not convinced that doing multiple statistical tests of the same data provides better information or more certainty for managers and policy makers. At the very least including this extra information makes the paper longer, and harder for a reader to understand and I don’t think that the added value is really there. I think the paper would be better with a single type of analysis or with additional analyses relegated to the supplement.

Minor issues:

• Line 1: The use of the phrase “The paradox” doesn’t feel like it is lived up to by the body of the manuscript. I think the phrase sets an expectation that 1.) This is a commonly observed phenomena, and 2.) You provide evidence that explains how this happens. I suggest a simpler title: “No in-stream algal response to forest and riparian zone management despite increases in light and nitrate”

• Line 39-41: I don’t really understand why the community being dominated by diatoms would explain the results.

• Line 41: Is the word “dynamics” needed here?

• Line 59: The topic sentence suggests that this is a list of ecosystem services provided by riparian zones. Ecosystem services are the benefits to humans provided by an ecosystem. Much of this list is benefits of riparian zones to streams which are only loosely tied back to humans in the last sentence of this paragraph. I suggest dropping the phrase “ecosystem services” from this paragraph.

• Line 31: You use the word “paradigm” in the abstract to describe the expected increase in nutrients, light, and shift towards autotrophy. However, on lines 79-80, and lines 87-89 you list numerous studies have found contrary findings. If there are already more than half a dozen studies that contradicted what you describe as the current “paradigm” I am not sure it is correct to describe it as a paradigm at all.

• Line 96: I am not sure what the phrase “dueling statistics” means.

• Line 104-117: The formatting of questions and hypotheses could be streamlined.

• Line 312: When you say “various measurements” in this sentence I don’t know if you mean measurements of different parameters within the same stream and date, or a single parameter among dates and streams.

• Lines 485-486: I think there is an extra word in this sentence.

• Line 502-504: Including self-citation of unpublished work is not necessary to make this point.

• Line 519-522: This should be modified slightly. The buffers could have taken up some nitrogen. Perhaps modify this sentence to say “Although not all transport of inorganic nitrogen…”

• Line 499-502: Forest harvest increased nitrogen concentrations in both nitrogen limited and non-nitrogen limited ecosystems. If the response is the same why draw a distinction between nitrogen limited and non-limited?

• When working with your data I had to refer back to the manuscript to figure out which sites had which treatment. It would be helpful if there were columns indicating treatments, and treatment periods in the data.

Reviewer #2: The manuscript addresses a relevant issue about limiting primary productivity in headwater streams, often limited by light, facing deforestation. It has an exciting approach, including experimental deforestation and long-term monitoring, potentially responding to relevant ecological questions, and dissertating about environmental management.

I noticed a potential issue concerning the work's message that seems dissonant with the research findings—for example, assessing the standing stock of biomass or pigments is not a direct measurement of primary productivity. Net primary production is strictly defined as the difference between the energy fixed by autotrophs and their respiration, and it is most commonly equated to increments in biomass per unit of surface and time, e.g., g C m-2 d-1 (https://www.sciencedirect.com/topics/earth-and-planetary-sciences/net-primary-production).

Could authors calculate NPP based on epileptic mass over time?

NPP may not be associated with biomass or pigments. For example, epilithon/periphyton is a matrix formed by a stock of autotrophic and heterotrophic biomass. Whether the heterotrophic compartment decrease and the autotrophic one increases, the balance is no biomass gain, and other factors may influence autotrophic and heterotrophic proportion (Guariento et al., 2011, https://link.springer.com/article/10.1007/s10452-011-9377-5). In addition, it is known that primary producers can increase or decrease their cell proportion of pigments depending on light conditions (Ma et al., 2022, https://doi.org/10.1016/j.biortech.2022.126777).

It is worth mentioning that monitoring data is precious, including the immediate and long-term impacts, and this research could show interesting time series. The way authors present 10 years of data pulled in before and after deforestation, using repeated measurements and ignoring system dynamics.

In addition, I believe that the manuscript should be restructured textually. For example, part of the assumptions is in the methods, and part of the methods is in the introduction. I find in the scientific literature a structure in which the introduction takes the scientific question and presents the hypotheses and the expected results. I also suggest including a sub-item about management and relocating textual sections accordingly.

Similarly to the first comment, it is unclear what processes were changed or not during the removal of the forest. What ecological processes and functions lead to changes and maintenance in the system? Is there evidence for what limitations of local N and P? Does species composition show any exciting pattern?

Finally, I believe the authors can tailor their questions to the results and respond to other ecological questions not addressed in this manuscript, such as biomass and species dynamics.

Reviewer #3: Thank you for giving me the opportunity to review this paper submitted to PLOS One. I think that the dataset analyzed in this paper has many strengths, including the longevity of the research with multiple years of pre-harvest controls, numerous watershed pairings, consideration of spatial complexities when studying stream ecosystems, extensive complementary datasets from a multi-disciplinary team, consistent annual sampling design, and algal composition data reviewed by a co-author who leads the field in diatom identification. While I could not speak specifically to the nuances of the partial least squares regression method due to my lack of familiarity with this type of analysis, I recognize the strength of using multiple non-parametric statistical methods to analyze field data characterized by a high degree of natural variability. Consequently, these data offer an important contribution to our understanding of how forest management affects in-stream primary producers and I sincerely hope that these data are published. However, I did feel that there were numerous and substantial issues with the paper as written that lead me to recommend that this paper be categorized as needing “major revisions.” My concerns with the paper tend to fall into three categories. First, while there were many strengths in the methodologies used in this study, I do not feel that the authors provided enough detail on the specifics of how algae were collected from the stream in the methods and did not do enough to recognize the limitations of some of the methods that they used in the discussion. Second, while many of the figures were relatively visually clear, I found some inconsistencies with labeling to make it difficult to compare among figures. There was also a substantial amount of information that I felt was missing from the figure legends pertaining to the statistical analyses. I also disagreed with the use of standard deviations for error bars when the authors calculated 95% confidence intervals as part of their statistical analyses (and are more easily interpreted for statistical relevance in bar graphs). Finally, I felt that there were some substantial issues with the writing, particularly related to the framing of the data analysis and the literature used to support their findings. In particular, there were many areas of the writing that were too broad and drew from a broader literature source; focusing more specifically on riparian studies in headwater streams would allow the authors to make more direct comparisons and strengthen their claims.

Please see the attached document for specific in-line comments for the authors.

6. PLOS authors have the option to publish the peer review history of their article (what does this mean?). If published, this will include your full peer review and any attached files.

Reviewer #1: No

Reviewer #2: **Yes: **Fausto Machado-Silva

Reviewer #3: No

---

## [Author Response · Author response to Decision Letter 0]

22 Dec 2022

I uploaded our response to reviewer and editor comments as a separate document. It is 25 pages of text, so best viewed as a document. I also addressed the editor comment about funding sources in our updated cover letter.

---

## [Decision Letter · Decision Letter 1]

28 Feb 2023

PONE-D-22-25727R1Stream nitrate enrichment and increased light yet no algal response following forest harvest and experimental manipulation of headwater riparian zonesPLOS ONE

Dear Dr. Johnson,

Thank you for submitting your manuscript to PLOS ONE. After careful consideration, we feel that it has merit but does not fully meet PLOS ONE’s publication criteria as it currently stands. Therefore, we invite you to submit a revised version of the manuscript that addresses the points raised during the review process.

We look forward to receiving your revised manuscript.

Kind regards,

Clara Mendoza-Lera

Academic Editor

PLOS ONE

Journal Requirements:

Reviewers' comments:

Reviewer's Responses to Questions

**Comments to the Author**

1. If the authors have adequately addressed your comments raised in a previous round of review and you feel that this manuscript is now acceptable for publication, you may indicate that here to bypass the “Comments to the Author” section, enter your conflict of interest statement in the “Confidential to Editor” section, and submit your "Accept" recommendation.

Reviewer #4: All comments have been addressed

2. Is the manuscript technically sound, and do the data support the conclusions?

Reviewer #4: Yes

3. Has the statistical analysis been performed appropriately and rigorously? 

Reviewer #4: Yes

4. Have the authors made all data underlying the findings in their manuscript fully available?

Reviewer #4: Yes

5. Is the manuscript presented in an intelligible fashion and written in standard English?

Reviewer #4: Yes

6. Review Comments to the Author

Reviewer #4: Dear authors.

The manuscript titled “Stream nitrate enrichment and increased light yet no algal response following forest harvest and experimental manipulation of headwater riparian zones” aims to analyze changes in epilithic biofilm, chlorophyll a, nutrients, and sediment deposition in headwater streams under pre and post-harvest comparing with reference sites and analyzing the effects in downstream sites on the watershed. Although the researchers found a significant increase in light and nutrients (in this case DIN), no significant differences were found on the tested epilithic biofilm and chlorophyll a, possibly related to a co-limitation driven by the low phosphorous concentration.

It seems to me that the statistical approaches are quite accurate to try to find the relationships at different levels, either treating the watersheds as treatments or analyzing each one given the great variability that each watershed has. That is why it seems to me that the responses of the authors to the previous reviews make it clear why the use of multiple statistical analyzes is adequate for this type of situation and I agree that the authors mention that the lack of response from the variables mentioned to environmental changes is not always reported in previous studies. This gives great importance to this article being published since it can serve as a starting point for the following researchers that can give a more powerful response to decision-makers in the management and management of hydrobiological resources.

Solo adjunto algunos errores tipográficos en los manuscritos .pfd.

7. PLOS authors have the option to publish the peer review history of their article (what does this mean?). If published, this will include your full peer review and any attached files.

Reviewer #4: No

---

## [Author Response · Author response to Decision Letter 1]

30 Mar 2023

We uploaded our Response to Reviewer document. In it, we fully complied with the small number of suggested edits and comments

---

## [Editor Report · Decision Letter 2]

4 Apr 2023

Stream nitrate enrichment and increased light yet no algal response following forest harvest and experimental manipulation of headwater riparian zones

PONE-D-22-25727R2

Dear Dr. Johnson,

We’re pleased to inform you that your manuscript has been judged scientifically suitable for publication and will be formally accepted for publication once it meets all outstanding technical requirements.

Kind regards,

Clara Mendoza-Lera

Academic Editor

PLOS ONE
---

## [Editor Report · Acceptance letter]

11 Apr 2023

PONE-D-22-25727R2 

Stream nitrate enrichment and increased light yet no algal response following forest harvest and experimental manipulation of headwater riparian zones 

Dear Dr. Johnson:

I'm pleased to inform you that your manuscript has been deemed suitable for publication in PLOS ONE. Congratulations! Your manuscript is now with our production department. 

Kind regards, 

on behalf of

Dr. Clara Mendoza-Lera 

Academic Editor

PLOS ONE